# A novel N-terminal extension in mitochondrial TRAP1 serves as a thermal regulator of chaperone activity

James R Partridge[1†‡], Laura A Lavery[1†§], Daniel Elnatan[1], Nariman Naber[2], Roger Cooke[2], David A Agard[1]*

[1]Department of Biochemistry and Biophysics, Howard Hughes Medical Institute, University of California, San Francisco, San Francisco, United States; [2]Department of Biochemistry and Biophysics, University of California, San Francisco, San Francisco, United States

*For correspondence: agard@msg.ucsf.edu

†These authors contributed equally to this work

Present address: ‡Global Blood Therapeutics, South San Francisco, United States; §Baylor College of Medicine Departments of Molecular and Human Genetics, Jan and Dan Duncan Neurological Research Institute at Texas Children's Hospital, Houston, United States

Competing interests: The authors declare that no competing interests exist.

**Abstract** Hsp90 is a conserved chaperone that facilitates protein homeostasis. Our crystal structure of the mitochondrial Hsp90, TRAP1, revealed an extension of the N-terminal β-strand previously shown to cross between protomers in the closed state. In this study, we address the regulatory function of this extension or 'strap' and demonstrate its responsibility for an unusual temperature dependence in ATPase rates. This dependence is a consequence of a thermally sensitive kinetic barrier between the apo 'open' and ATP-bound 'closed' conformations. The strap stabilizes the closed state through trans-protomer interactions. Displacement of cis-protomer contacts from the apo state is rate-limiting for closure and ATP hydrolysis. Strap release is coupled to rotation of the N-terminal domain and dynamics of the nucleotide binding pocket lid. The strap is conserved in higher eukaryotes but absent from yeast and prokaryotes suggesting its role as a thermal and kinetic regulator, adapting Hsp90s to the demands of unique cellular and organismal environments.

## Introduction

Hsp90 is a highly conserved molecular chaperone essential for protein and cellular homeostasis. Although molecular chaperones generally promote protein folding and prevent aggregation, Hsp90 is unique in that it interacts with substrate ('client') proteins that are already in a semi-folded state to facilitate downstream protein–protein interactions and promote client function in diverse biological pathways (*Jakob et al., 1995*; *Taipale et al., 2012*). Hsp90 interacts with nearly 10% of the eukaryotic proteome (*Zhao et al., 2005*), and its client proteins vary significantly in sequence, structure, and size (*Echeverria et al., 2011*). In most eukaryotes, there are four different Hsp90 homologs: Hsp90α and Hsp90β in the cytoplasm, Grp94 in the endoplasmic reticulum (ER), and TRAP1 in mitochondria, with each homolog contributing unique biological functions (*Chen et al., 2006*; *Johnson, 2012*). Deregulation of Hsp90 protein levels and function has been linked to multiple human diseases and for this reason Hsp90 is a target for biochemical characterization, structural studies, and drug discovery (*Luo et al., 2010*; *Taipale et al., 2010*). Despite such importance, little is known about the biochemical characteristics that regulate client interaction and specificity.

Hsp90 exists as a homodimer, with each protomer consisting of three major domains. The N-terminal domain (NTD) binds to ATP, the C-terminal domain (CTD) provides a dimerization interface between protomers, and the middle domain (MD) provides a stabilizing γ-phosphate contact to help facilitate ATP hydrolysis (*Cunningham et al., 2012*). Together with the CTD, the MD has been shown to aid in

**eLife digest** Proteins—which are made of chains of molecules called amino acids—play many important roles in cells. Before a newly made protein can work properly, the amino acid chain has to be folded into the correct three-dimensional shape. Many proteins that have folded incorrectly are harmless, but some can disrupt the cell and cause damage. Although most proteins can fold properly on their own, they are often helped by 'chaperone' proteins, which speed up the process and encourage correct folding.

Many chaperone proteins belong to a family called the heat shock proteins, which are found in almost all species: from bacteria, to plants and animals. High temperatures can severely impair and destabilize proper protein folding, and the heat shock proteins counteract this by helping to prevent, or correct, protein misfolding. Most animals and plants have at least four genes that make different versions of heat shock protein 90 (Hsp90). These versions work in different places in the cell and one—called TRAP1—is found in internal compartments called mitochondria. Along with its role in assisting protein folding, TRAP1 also acts as an indicator of the health of the proteins in the mitochondria.

One section or 'domain' of Hsp90 is able to bind to and break down a molecule called ATP. This releases energy that is used to change the shape of the protein-binding domain—which is responsible for helping other proteins to fold. Recent studies of TRAP1 using a technique called protein crystallography highlighted the presence of a short amino acid tail or 'strap' at one end of the protein, but it is not known what role it may play in protein folding.

In this study, Partridge et al. reveal that the amino acid strap of TRAP1 controls the breakdown of ATP in a way that depends on the surrounding temperature. Similar straps are also present in the Hsp90 proteins that are found in other parts of the cell. However, the strap is absent from the Hsp90 proteins of yeast and bacteria. These experiments used proteins that had been taken from living cells and placed in an artificial setting, so an important next step will be to study the role of the strap in the folding of proteins inside living cells. Also, future work could investigate the potential role of the protein in maintaining healthy mitochondria.

the formation of client interactions (**Street et al., 2011**, **2012**; **Genest et al., 2013**). Large, rigid body motions about each of the domain interfaces give rise to an ensemble of remarkably diverse conformational states that dictate the functional Hsp90 cycle (**Ali et al., 2006**; **Shiau et al., 2006**; **Dollins et al., 2007**; **Southworth and Agard, 2008**; **Lavery et al., 2014**) (**Video 1**) and are linked to client maturation in vivo (**Panaretou et al., 1998**). Work from numerous labs has demonstrated conservation of the underlying conformational cycle and mechanism; however, each Hsp90 homolog has a distinct conformational equilibrium and catalytic rate (**Panaretou et al., 1998**; **Richter et al., 2008**; **Southworth and Agard, 2008**). Binding of ATP to the NTD nucleotide-binding pocket ultimately leads to stabilization of an NTD-dimerized state. Key steps in this transformation include ATP binding, closure of a mobile structure (lid) over the nucleotide, and a subsequent 90° rotation of the NTD relative to the MD (**Krukenberg et al., 2011**). Dimer closure is the rate-limiting step for Hsp90 ATPase activity and mutations that either subtly increase or decrease ATPase rates compromise viability in yeast (**Nathan and Lindquist, 1995**; **Hessling et al., 2009**). However, our understanding of the sequence of events that regulate these structural rearrangements is limited.

Recently, we solved a series of full-length crystal structures of TRAP1 bound to different ATP analogs (**Lavery et al., 2014**), providing new insights into the structure, dynamics, and mechanism of Hsp90. Of particular note was the marked asymmetry between protomers of the homodimer, primarily at the interface between the MD and CTD. This asymmetry was sampled in solution, proved essential for catalytic turnover, and provided a new model for coupling the energy of ATP hydrolysis to client remodeling. A second feature highlighted by the TRAP1 crystal structure was an ordered 14-residue extension (out of 26 total additional residues) of the N-terminal β-strand previously shown to cross over ('swap') between protomers in the closed state (**Ali et al., 2006**). While absent in yeast and bacteria, this extension, or 'strap,' is found in most eukaryotic Hsp90 proteins including the cytosolic and organellar forms (**Chen et al., 2006**) and can extend for as many as 122 residues as recently found in a splice variant of Hsp90α in higher eukaryotes (**Tripathi and Obermann, 2013**). Structure based

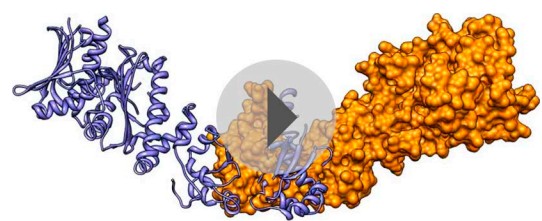

**Video 1.** Conformational dynamics of the Hsp90 cycle. A morph between known conformations throughout the activity cycle of Hsp90 (PDB codes with no order dictated: 2O1V, 2CG9, 2IOP, 2IOQ, 4IPE, 4IVG).

point mutations and complete removal of the TRAP1 strap (Δstrap) resulted in a sixfold increase in ATPase activity in zebrafish TRAP1 (zTRAP1) (*Lavery et al., 2014*), evidence that the strap plays a regulatory role. Similarly, deletion of the strap in Grp94 (residues 22-72, referred to as the 'pre-N domain') resulted in a fivefold increase in ATPase (*Dollins et al., 2007*), indicative of a conserved regulatory role, although the mechanism remains unclear.

In this study, we explore the conformational cycle of TRAP1 and demonstrate that the strap is responsible for a large thermal barrier between the apo (open) and ATP bound (closed) states. Using negative-stain electron microscopy (EM) and Small-Angle X-ray Scattering (SAXS), we demonstrate that removal of the strap results in a profound reduction in the temperature sensitivity observed in multiple TRAP1 homologs, indicating that the strap is responsible for this unique behavior. Additionally, we develop fluorescence resonance energy transfer (FRET) and continuous-wave EPR (CW-EPR) assays to show that the strap regulates the rate-limiting conformational transitions that precede NTD dimerization, including NTD rotation and lid closure over the ATP-binding pocket. These results indicate that the strap must stabilize both the apo state and the closed state, providing a unique evolutionary strategy for modulating different phases of the kinetic landscape and optimizing in vivo function of diverse Hsp90s.

## Results

### A temperature-sensitive kinetic barrier limits the conformational transition from apo to the closed state in TRAP1

With all previously studied Hsp90s, incubation with slowly- or non-hydrolyzable ATP analogs favors accumulation of a closed, NTD-dimerized state. However, the extent of closed-state accumulated and the rate of closure differentiated the Hsp90s with the individual values positively correlating with the ATP hydrolysis rate (*Richter et al., 2008*; *Southworth and Agard, 2008*; *Hessling et al., 2009*). Specifically, we used EM to demonstrate that the large variability in observed ATPase rates of cytosolic bacterial (bHsp90), yeast (yHsp90), and human Hsp90 (hHsp90), directly correlated with the ability of each homolog to reach a closed conformation in the presence of non-hydrolyzable ATP (AMPPNP) (*Southworth and Agard, 2008*). Here, negative-stain EM was again used to monitor the ability of human TRAP1 (hTRAP1) to transition from the apo conformation to the closed conformation in the presence of AMPPNP. While hTRAP1 has an ATPase rate similar to the *E. coli* Hsp90 (~0.5 min$^{-1}$) (*Cunningham et al., 2012*), surprisingly hTRAP1 remained in the open conformation despite incubation with saturating AMPPNP (*Figure 1A*). Noting that the discrepancy might be related to different incubation temperatures between the two experiments, we monitored the ability of hTRAP1:AMPPNP to close as a function of temperature. After 1 hr (*Figure 1A*) or overnight (*Figure 1B*) incubation at room temperature (RT, ~23°C) hTRAP1 remained in the open state. However, after a single hour of incubation at increasing temperatures, the closed state was increasingly populated (*Figure 1A*). These results correlate well with the temperature sensitive steady-state hydrolysis rates of hTRAP1 that increases by nearly 200-fold between 25°C and 55°C (*Leskovar et al., 2008*) and are consistent with closure being rate-limiting for hydrolysis. Importantly, the equilibrium reached at each temperature (*Figure 1A*) remains fixed after subsequent incubation at RT overnight (*Figure 1B*). These data suggest both a large, unusually thermally sensitive kinetic barrier to closure and a highly stable closed state.

### The N-terminal strap is responsible for TRAP1 thermal sensitivity

To better measure the equilibrium between conformational states as a function of temperature, we used SAXS, which can directly quantify the solution distribution of open and closed states (*Krukenberg et al., 2008*). As demonstrated by a shift towards a more compact pair-wise inter-atomic distance distribution, P(r), there was a strong correlation between temperature and dimer closure, (*Figure 2A*). By fitting

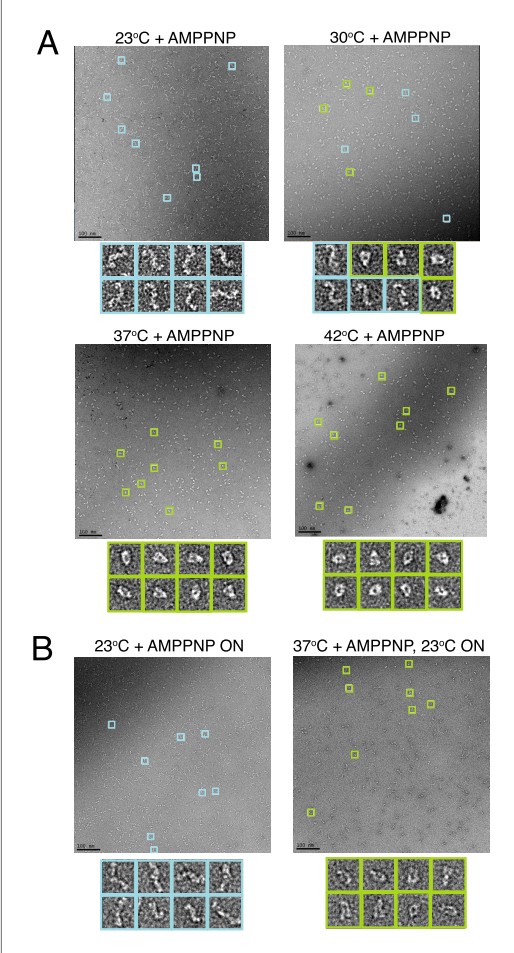

**Figure 1**. A temperature-dependent barrier separates the apo and closed state of TRAP1. (**A**) Negative stain electron microscopy (EM) images of hTRAP1 in the presence of AMPPNP at increasing temperatures for 1 hr. While the population at equilibrium appears to remain in an apo conformation at room temperature (RT), conversion to the closed state appears to be intermediate at 30°C and nearly complete at 37°C and 42°C. (**B**) Negative stain EM images of reactions incubated at 23°C and 37°C from **A** after returning the sample to RT and incubating overnight. Both populations remain apo and closed (respectively) demonstrating the large kinetic barrier that limits the conformational transition from apo to the closed state. Scale bar is 100 nm.

the distributions as a linear combination of open and closed states, the fraction of closed state can be accurately estimated ('Materials and methods'). After 1 hr at 20°C only 0.4% of the molecules have reached the closed conformation, while at 43°C roughly 84% of the molecules are closed (*Figure 2C* and *Table 1*). In agreement with our EM data, the equilibrium does not revert back to the apo state when the temperature is lowered (*Figure 2D*). Interestingly, TRAP1 from zebrafish (zTRAP1) displays a shifted temperature-dependent conformational equilibrium that correlates with its higher basal ATPase rate (*Figure 2* and *Table 2*) and the lower physiological temperature of zebrafish (~29°C).

Previous studies by Richter et al. had demonstrated that removal of the initial β-strand in yHsp90 (corresponding to post-strap residues in TRAP1) increased ATPase activity and facilitated N-terminal dimerization (*Frey et al., 2007*). The ordered strap extension observed in the zTRAP1 structure is also kinetically important, as Δstrap (deletion of zTRAP1 residues 73–100) and a single point mutant aimed at disrupting a conserved, stabilizing salt bridge at the beginning of the strap, accelerated hydrolysis by six-fold and fourfold, respectively. Together, these raised the possibility that the strap might also be responsible for the unusual temperature-regulated energy landscape observed in TRAP1 homologs.

As a first step, we show that in hTRAP1, strap removal (lacking residues 60–85) has an even more profound impact on ATPase activity (~30-fold) than on zTRAP1 (*Figure 3*, *Table 2*). The larger increase in ATPase activity for hTRAP1 correlates with the more significant temperature dependence (*Figure 2C*) and thus a higher kinetic barrier for hTRAP1 at the experimental temperature of 30°C. Notably, a smaller truncation lacking residues 60–69, that preserved the conserved His71:Glu142 salt-bridge in hTRAP1, did not have an effect on ATPase activity (*Figure 3*, *Table 2*).

Strikingly, SAXS revealed that at every temperature examined in both hTRAP1- and zTRAP1-Δstrap immediately equilibrated to a closed conformation after adding AMPPNP, indicating a loss of temperature sensitivity (at least at temperatures ≥20°C) (*Figure 2B*). This indicates that beyond a role in stabilizing the closed conformation through trans-protomer interactions, the strap must also be involved in apo interactions that inhibit a transition towards the closed state. These data together with previously solved crystal structures of other Hsp90 N-terminal domains displaying cis-contacts of the initial β-strand suggest that the strap likely makes equivalent contacts with the same NTD (cis) in the apo state that it forms with the trans-NTD in the closed state (*Shiau et al., 2006*; *Dollins et al., 2007*; *Li et al., 2012*).

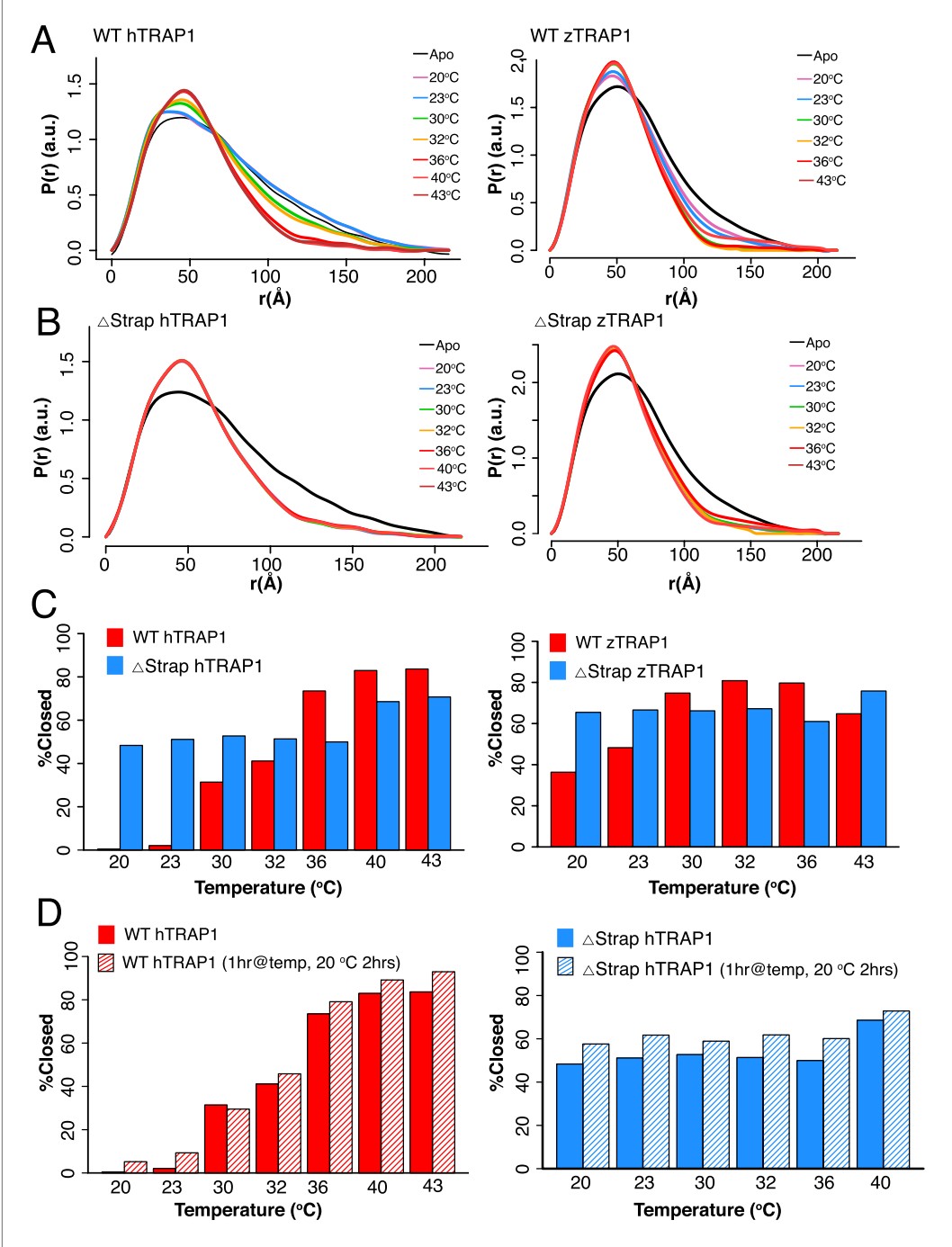

**Figure 2**. A large energy barrier to the closed state is modulated by the NTD-strap. (**A**) SAXS distributions at equilibrium for hTRAP1 (left) and zTRAP1 (right) (84% identical to hTRAP1) in apo and in the presence of saturating AMPPNP at indicated temperatures for 1 hr. The closed-state population substantially increases at and above 36°C for hTRAP1 while zTRAP1 maintains a higher level of % closed at even lower temperatures, consistent with the differences in physiological temperatures of the two species. (**B**) SAXS distributions of Δstrap in matching conditions from **A** showing that removal of the strap mitigates the temperature-dependent barrier between the apo and closed states. (**C**) Quantification of percent closed for both TRAP1 species ± the strap region. Apparent is the different temperature dependence of hTRAP1 and zTRAP1 and the loss of temperature response of the chaperone in the case of Δstrap. (**D**) A plot of percent closed state verses temperature of WT hTRAP1 (left) and Δstrap hTRAP1 after closure has completed at each given temperature (solid bars as in (**C**). These samples were then cooled for 2 hr at 20°C (stripped bars). The data suggest a highly stable closed state.

**Table 1.** Quantification of percent closed using SAXS data for both TRAP1 species ± the strap

| Temperature (°C) | Protein | % Closed state | R |
|---|---|---|---|
| 20 | WT hTRAP1 | 0.5 | 0.042 |
| 23 | WT hTRAP1 | 2 | 0.042 |
| 30 | WT hTRAP1 | 31 | 0.024 |
| 32 | WT hTRAP1 | 41 | 0.019 |
| 36 | WT hTRAP1 | 74 | 0.011 |
| 40 | WT hTRAP1 | 83 | 0.010 |
| 43 | WT hTRAP1 | 84 | 0.012 |
| 20 | WT zTRAP1 | 36 | 0.015 |
| 23 | WT zTRAP1 | 48 | 0.011 |
| 30 | WT zTRAP1 | 75 | 0.027 |
| 32 | WT zTRAP1 | 81 | 0.033 |
| 36 | WT zTRAP1 | 80 | 0.030 |
| 43 | WT zTRAP1 | 69 | 0.016 |
| 20 | hTRAP1 Δstrap | 66 | 0.015 |
| 23 | hTRAP1 Δstrap | 68 | 0.014 |
| 30 | hTRAP1 Δstrap | 69 | 0.014 |
| 32 | hTRAP1 Δstrap | 68 | 0.015 |
| 36 | hTRAP1 Δstrap | 67 | 0.018 |
| 40 | hTRAP1 Δstrap | 69 | 0.016 |
| 43 | hTRAP1 Δstrap | 71 | 0.015 |
| 20 | zTRAP1 Δstrap | 60 | 0.016 |
| 23 | zTRAP1 Δstrap | 64 | 0.014 |
| 30 | zTRAP1 Δstrap | 61 | 0.015 |
| 32 | zTRAP1 Δstrap | 62 | 0.014 |
| 36 | zTRAP1 Δstrap | 55 | 0.016 |
| 43 | zTRAP1 Δstrap | 76 | 0.011 |

## NTD-strap limits closure rate by regulating NTD rotation and lid dynamics

The above results predict that the rate of closure should be proportional to temperature. Fluorescence Resonance Energy Transfer (FRET) provides a more convenient method than SAXS to directly measure the rate of closure (*Hessling et al., 2009*; *Mickler et al., 2009*; *Street et al., 2011*). Two FRET constructs were designed to probe distinct aspects of the closure reaction, relying on a Cys-free version of hTRAP1 (*Lavery et al., 2014*). The first construct modeled from FRET positions previously designed for yHsp90 placed a single Cys residue on each protomer (E140C and E407C, 'Inter FRET') so as to give an increase in FRET upon closure (*Hessling et al., 2009*). The second construct modeled on previous work with bHsp90 (*Street et al., 2012*), adds two Cys residues to a single protomer (S133C and E407C, 'Intra FRET'), and is designed to track the ~90° NTD rotation (relative to the MD) that occurs upon closure. After forming heterodimers, closure reactions were initiated with AMPPNP over a temperature range mirroring our SAXS experiments and the change in FRET was monitored. Pre- and post-reaction fluorescent scans showed a predicted FRET change indicative of closure for each FRET construct (*Figure 4A*). As expected, the rate of closure correlated with increasing temperature (*Figure 4B*, *Figure 4—figure supplement 1*, *Table 3*) for both dimer closure and NTD:MD rotation measurements. To measure the contribution of the strap to the kinetics of closure, we truncated the strap region of either one or both protomers in each FRET construct (although the dimeric Δstrap construct used to measure NTD rotation proved too unstable to obtain reliable data). In both cases, a large acceleration of closure was apparent (*Figure 4C*) with the largest acceleration (16-fold) observed for the double-strap deletion.

A good way to quantitate the contribution of the strap to the thermal barrier is to measure closure rates as a function of temperature with and without the strap (*Figure 4B,D* and *Figure 4—figure supplement 1*) and to calculate the activation energy ($E_a$) towards closure (i.e., the temperature dependent barrier height). At every temperature sampled removing the strap results in an acceleration of closure compared to WT and an overall loss in temperature dependence (*Figure 4D*). Comparing the fold changes in closure rates (*Table 3*), we see the largest fold change at lower temperatures (23°C: 24-fold, 30°C: 16-fold, 32°C: 12-fold, 36°C: sevenfold, and 42°C: threefold). This increased impact at lower temperatures is readily evident in an Arrhenius plot calculated from the inter FRET experiments (*Figure 4E*). The resultant activation energies ($E_a$) taken from the slopes of these curves are 48.8 kcal/mol and 29 kcal/mol, for WT and Δstrap respectively. From the difference, the strap appears to be contributing ~20 kcal/mol towards the $E_a$ of WT hTRAP1, which we interpret as ~ 20 kcal/mol of enthalpic stabilization of the open state. Our $E_a$ for WT hTRAP1 is consistent with that measured previously under slightly different conditions (*Leskovar et al., 2008*), but is considerably higher than that calculated for other Hsp90 homologs (*Figure 4—figure supplement 2*) (*Frey et al., 2007*). As a control, we also measured steady-state ATPase rates on the labeled protein used for the FRET

**Table 2.** Steady-state ATP hydrolysis rates at temperatures and buffer conditions of assay specified (i.e., EPR is under EPR buffer and temperature conditions). If not noted (top four reactions), conditions are the same as reference (*Lavery et al., 2014*)

| Protein | zTRAP1 ATPase (min⁻¹) | hTRAP1 ATPase (min⁻¹) |
|---|---|---|
| WT (30°C) | 1.36 ± 0.12 | 0.463 ± 0.003 |
| salt bridge point mutants (30°C) (*Lavery et al., 2014*) | (E-A) 3.57 ± 0.62 (H-A) 5.08 ± 0.90 | |
| Δstrap (30°C) | 5.84 ± 0.47 | 13.3 ± 0.5 |
| Δ60-69 (30°C) | | 0.47 ± 0.02 |
| WT FRET (30°C) | | 0.21 ± 0.0.01 |
| Δstrap FRET (30°C) | | 11.9 ± 2.1 |
| CFree WT EPR (23°C) | 0.88 ± 0.05 | |
| CFree Δstrap EPR (23°C) | 5.65 ± 0.22 | |
| CFree WT (Inter FRET) (30°C) | | 0.79 ± 0.0.03 |
| CFree Δstrap (Inter FRET) (30°C) | | 7.6 ± 0.36 |
| CFree WT (Intra FRET) (30°C) | | 0.35 ± 0.0.01 |

Red text indicates WT or strap truncated protein with native cysteine and label free, while Blue indicates labeled protein used in FRET and EPR experiments (each in indicated buffer conditions). EPR samples are cysteine free except for the desired probe position and are spin-labeled. Inter FRET and Intra FRET samples are cysteine free except for the desired probe position and are labeled with Alexa Fluor dyes (Life Technologies, see 'Materials and methods'). Note that 'Intra FRET' refers to both probe positions on the same promoter, whereas 'Inter FRET' refers to one probe position per promoter. Errors represent the standard deviation of triplicate experiments.

experiments. While these showed differences in absolute ATPase rates between 1.5 and fourfold compared with their unlabeled counterparts (*Tables 2 and 3*), the relative impact of strap deletion was consistent across experiments. Together, these data support a model in which the N-terminal strap limits closure by inhibiting the rotational movement of the NTD that is necessary to form the catalytically active closed state.

To probe the underlying mechanism of the NTD-strap in the closure reaction, we sought to examine the relationship of the strap to the dynamics of the NTD lid (zTRAP1 residues 191–217) that closes over the ATP binding pocket; a mechanism conserved in many ATPases. Previous studies with yHsp90 have suggested a correlation between the 'β-strand swap' and dynamics of the NTD lid (*Richter et al., 2006*). In an open conformation and prior to nucleotide binding, the lid makes contacts with helix 1 (H1) (*Richter et al., 2006*; *Shiau et al., 2006*; *Dollins et al., 2007*; *Li et al., 2012*), while in the closed state the lid rotates to secure nucleotide via interactions at conserved sidechains (Ser193 and Ser195 in zTRAP1) inside the nucleotide binding pocket (*Ali et al., 2006*; *Lavery et al., 2014*) (*Videos 2 and 3*). This closed state lid conformation is incompatible with the NTD:MD apo state conformation as it would clash with the MD (*Shiau et al., 2006*; *Dollins et al., 2007*).

To test whether the strap has a role in lid stabilization, we developed an electron paramagnetic resonance (EPR) spectroscopy assay to track lid mobility in the apo and closed states ('Materials and methods'). A cys-free version of zTRAP1 with an Ala201Cys mutation allowed labeling of a fully accessible cysteine residue in the lid with N-(1-oxyl- 2,2,6,6-tetramethyl-4-piperidinyl)maleimide (MSL). We observed a small difference in ATPase activity with the MSL-labeled TRAP1 compared with ATPase rates measured using WT TRAP1 suggesting a minor labeling effect on steady-state catalytic turnover (~1.3 fold). EPR spectra are sensitive to the rotational mobility of the attached MSL probe making it a useful reporter for changes in local conformational dynamics (*Hubbell et al., 2000*). EPR spectra of full-length zTRAP1 were recorded at 23°C and shown to be more mobile in the nucleotide bound state compared to the apo state (*Figure 5A*). Mobility of the lid as measured with EPR is consistent with apo structures showing low B-factors in this region due to significant contacts with H1 of the cis protomer (*Richter et al., 2006*; *Shiau et al., 2006*). Conversely, crystal structures of TRAP1 and other Hsp90 homologs bound to ATP analogs in the closed and dimerized conformation show that

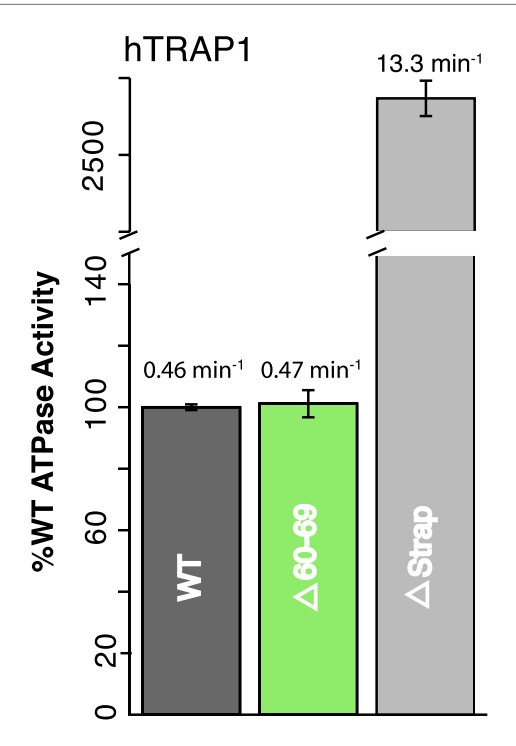

**Figure 3**. NTD-strap regulates ATP hydrolysis rates. WT and strap mutants for hTRAP1. Removal of the strap (Δstrap) results in a ~30-fold increase in ATPase rate, while truncations before the previously reported salt bridge contact Δ60–69 (*Lavery et al., 2014*) show no change in activity. Average steady-state hydrolysis rates (min$^{-1}$) above each bar, standard deviation of triplicate measurements can be found in *Table 2*.

the lid folds over the nucleotide, has increased B-factors and lacks many of the stabilizing contacts with the N-terminal domain found in the apo state (*Ali et al., 2006*; *Lavery et al., 2014*). This is consistent with the mobile signature in the EPR observed for the closed conformation. Comparing apo state equilibrium measurements for WT and Δstrap shows little change upon strap deletion (*Figure 5A*). Fortunately EPR is sufficiently sensitive and the closure kinetics for TRAP1 are sufficiently slow, that it is possible to directly monitor changes in lid state over time. By plotting the change in normalized peak heights over time ('Materials and methods', *Figure 5B*), it is apparent that the amplitude changes for both the mobile and immobile peaks are well fit by a single exponential curve for each sample. From this, it is clear that the rate of change between states as monitored by lid mobility is much faster for the Δstrap sample than for WT. The fold difference between rates is on the order of changes in ATPase rates under conditions used in the EPR experiment (*Table 2*). Altogether, these data suggest that the local conformational changes of lid closure and NTD-rotation are part of the rate-limiting barrier to the closed state and are regulated by N-terminal residues of the strand swap and extended strap in TRAP1.

## Dissecting further regulatory functions of the NTD-strap

The experiments above collectively suggest a major role for the N-terminal strap as a direct modulator of the kinetic barrier separating the apo and closed states for TRAP1. Moreover, it appears to be also responsible for the pronounced temperature-sensitivity. Although the experiments above indicate a strong role in modulating the forward closure rate, the TRAP1 crystal structure would suggest that deleting the strap might also compromise the stability of the closed state, thereby enhancing reopening rate and shifting the equilibrium towards the open state. To measure the re-opening rate, inter FRET-labeled hTRAP1 was pre-closed with AMPPNP. After closure was complete a 20-fold excess ADP was added such that upon re-opening of the NTD dimer interface, ADP would exchange resulting in a decreased FRET signal (*Street et al., 2011*). Previous studies found apo state nucleotide on and off-rates to be fast (*Leskovar et al., 2008*), thus the above experiment provides a good approximation of the uni-molecular reopening rate. Monitoring FRET kinetics revealed that strap removal accelerated re-opening of the NTD dimer interface by ~eightfold (0.0021 min$^{-1}$ → 0.016 min$^{-1}$; *Figure 6A*, *Table 3*). These data suggest that the strap contacts observed in the closed state (*Lavery et al., 2014*) do in fact impact closed-state stability, by about 1.2 kcal/mol, however, the larger effect (~16-fold, 0.02 min$^{-1}$ → 0.31 min$^{-1}$, 1.7 kcal/mol) is on the kinetic barrier corresponding to release of the strap from the apo state.

Because the strap could also play a role in the hydrolysis reaction, we needed a method to decouple closure from ATP hydrolysis. As closure is rate-limiting, even single-turnover experiments would provide an aggregate rate made up of the closure and hydrolysis steps. During the course of our FRET experiments, we discovered that omitting Mg$^{2+}$ from the reaction buffer results in an accumulation of the closed-state in the presence of ATP without ATP hydrolysis. By contrast, in the presence of ATP and Mg$^{2+}$, TRAP1 is predominantly in the apo state as a consequence of hydrolysis (*Figure 6B,D*). The latter is consistent with previous observations with yHsp90 (*Hessling et al., 2009*) and bHsp90. These

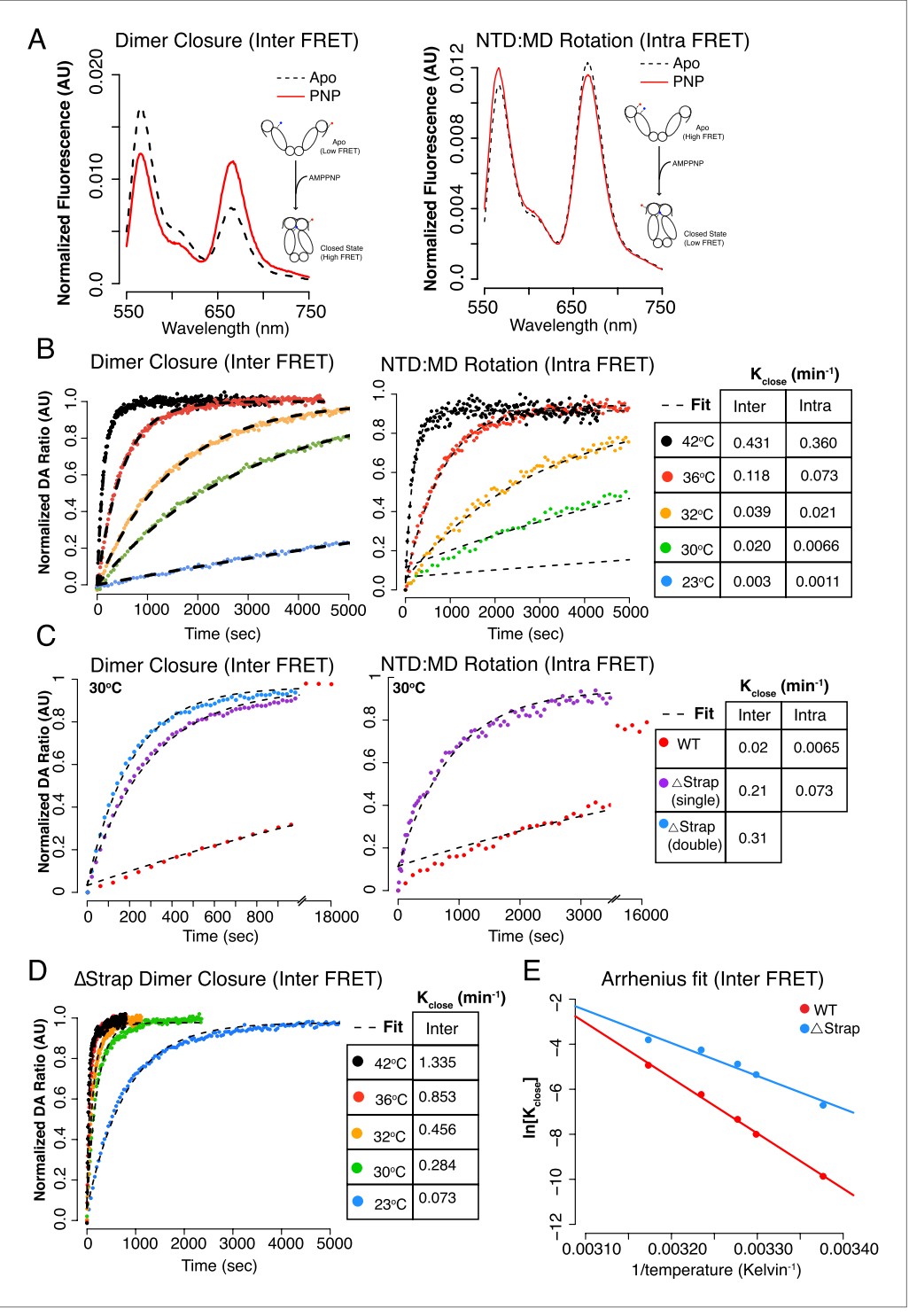

**Figure 4**. The NTD-strap regulates closure rate of TRAP1. (**A**) Steady-state FRET scans at 23°C for apo and AMPPNP reactions after closure with AMPPNP reached completion illustrating the anti-correlated change in FRET upon closure as measured by 'dimer closure' between protomers (left, Inter FRET) and rotation of the NTD from apo to the closed state within one protomer 'NTD:MD Rotation' (right, Intra FRET). (**B**) Temperature-dependent closure rates for WT hTRAP1 measured by both the dimer closure and NTD rotation FRET probes from **A**. Closure rates are comparable between these two sets of FRET probes as indicated in the table to the right. The predicted increase in rate at higher temperatures is apparent. (**C**) Closure at 30°C of WT compared to heterodimers lacking one or both NTD strap residues measured by dimer closure FRET (left) and NTD rotation FRET (right). Closure rates

*Figure 4. Continued on next page*

*Figure 4. Continued*

are found in the table for each experiment. (**D**) Temperature-dependent closure rates of Δstrap protein measured using the dimer closure probes from **A** (Inter FRET) illustrating both a rate acceleration and a dramatic loss of temperature dependence compared to WT (B, left panel). (**E**) Arrhenius plot of WT and Δstrap plotted using data from panels (**B**) (left) and (**D**). From the difference in activation energies $E_a$ between WT and Δstrap, the strap contributes approximately 60% of the measured $E_a$ for WT hTRAP1 (48.8 kcal/mol $E_a$ for WT; 29 kcal/mol Δstrap). These data are consistent with the steady-state SAXS and ATPase and show that removal of the strap region lowers the energy barrier between apo and the closed state.

The following figure supplements are available for figure 4:

**Figure supplement 1**. Alternative view of curve fits for *Figure 4B,D*.

**Figure supplement 2**. Arrhenius plots for Hsp90 homologs plotted using data from reference (*Frey et al., 2007*).

observations allowed us to decouple the closure and hydrolysis steps by pre-incubating TRAP1 with excess ATP without $Mg^{2+}$, thereby stalling the reaction in the closed state (illustrated in *Figure 6C*). Upon addition of $Mg^{2+}$, ATP is hydrolyzed and the equilibrium shifts predominantly to the apo state as seen by the loss of FRET (*Figure 6B*). Testing WT and Δstrap in this assay revealed an acceleration of closure with removal of the strap, consistent with experiments using AMPPNP (*Figure 6D*). Interestingly, the closure rate measured by FRET is significantly faster with ATP than AMPPNP suggesting a significant difference in energetics between the nucleotide analogs (*Table 3*). The use of ATP for FRET-based closure measurements better matches our ATPase measurements and points to a correlation between closure rates and ATPase activity (0.79 $min^{-1}$ ATPase vs 0.42 $min^{-1}$ FRET Closure, both measurements with Inter FRET probe protein), though we do still observe a difference perhaps representing a small $Mg^{2+}$ contribution. Addition of excess $Mg^{2+}$ showed the predicted drop in FRET and revealed a minor difference in hydrolysis rate (~1.4-fold) (*Figure 6E*, *Table 3*), suggesting that the strap may also subtly alter lid dynamics in the closed state. The acceleration effects observed for the Δstrap protein are greater at 25°C, where the temperature dependent difference is more pronounced (*Figure 6—figure supplement 1*, *Table 3*). Notably, our measured closure and hydrolysis rates matched previously reported values for these steps modeled using a global fitting procedure (*Leskovar et al., 2008*). However, the closure and hydrolysis rates measured here (0.003 $s^{-1}$ and 0.07 $s^{-1}$, respectively) were somewhat arbitrarily assigned to the reverse order in the previously reported model. Since our experiments independently measure both reactions, we can now assign closure to be the slowest and hence rate-limiting step. This model is in good agreement with the other data presented in this study.

Our combined data better define the kinetic cycle for TRAP1 and support a model where the strap regulates multiple steps with the largest contribution being to the thermal sensitive rate-limiting kinetic barrier between the apo and nucleotide-bound closed states.

## Discussion

The conservation of Hsp90 has been established from bacteria to humans, giving rise to homologs in different species and distinct versions in different cellular compartments (*Johnson, 2012*). Though biochemical and structural studies have identified key differences in the thermodynamic and kinetic properties amongst the homologs, the underlying set of conformations and the overall ATP hydrolysis cycle appear conserved and essential for client maturation in vivo (*Panaretou et al., 1998*; *Southworth and Agard, 2008*).

Here, we identify and characterize unique kinetic and thermodynamic properties of the mitochondrial Hsp90 (TRAP1) and use a combination of biophysical and biochemical techniques to consistently show that a 26-residue N-terminal extension or 'strap' (compared to yHsp90) (*Lavery et al., 2014*), kinetically regulates the formation of the active closed conformation and is responsible for the surprising temperature-dependence of closure. This extension is elaborated to varying degrees in the different Hsp90 isoforms; absent in yeast and bacterial Hsp90s, shortest in the dominantly expressed mammalian cytosolic Hsp90s and longest in the mammalian organellar Hsp90s (*Figure 7*). Below we propose that extensions and variability in the N-terminal sequence serve to fine-tune the activity of Hsp90 homologs in diverse species or compartments in response to functional demands and environmental factors, with temperature playing an important role in TRAP1.

**Table 3.** Kinetics of conformational changes as measured by FRET. Errors represent the standard deviation of triplicate experiments

| Protein | Temperature (°C) | FRET probe position | $K_{close}$ (min⁻¹) | $K_{reopen}$ (min⁻¹) | $K_{hyd}$ (min⁻¹) |
|---|---|---|---|---|---|
| CFree WT hTRAP1 (Intra FRET) | 23 | S133C.E407C | 0.0011 ± 0.00006 | | |
| | 30 | S133C.E407C | 0.0066 ± 0.00007 | | |
| | 32 | S133C.E407C | 0.021 ± 0.001 | | |
| | 36 | S133C.E407C | 0.073 ± 0.002 | | |
| | 42 | S133C.E407C | 0.36 ± 0.011 | | |
| | 30 | S133C.E407C | 0.073 ± 0.005 | | |
| CFree WT hTRAP1 (Inter FRET) | 23 | E140C/ E407C | 0.003 | | |
| | 30 | E140C/ E407C | 0.02 ± 0.002 | 0.00210 ± 0.00003 | |
| | 32 | E140C/ E407C | 0.039 | | |
| | 36 | E140C/ E407C | 0.118 | | |
| | 42 | E140C/ E407C | 0.431 | | |
| CFree Δstrap single (Inter FRET) | 30 | E140C/ E407C | 0.21 ± 0.013 | | |
| CFree Δstrap double (Inter FRET) | 23 | E140C/ E407C | 0.073 | | |
| | 30 | E140C/ E407C | 0.31 ± 0.024 | 0.016 ± 0.003 | |
| | 32 | E140C/ E407C | 0.456 | | |
| | 36 | E140C/ E407C | 0.853 | | |
| | 42 | E140C/ E407C | 1.335 | | |
| *CFree WT hTRAP1 (Inter FRET) *ATP used | 30 | E140C/ E407C | 0.42 ± 0.01 | | 7.56 ± 0.99 |
| | 25 | E140C/ E407C | 0.19 ± 0.01 (0.003 ± 0.0001 s⁻¹) | | 4.3 ± 0.2 (0.071 ± 0.004 s⁻¹) |
| *CFree Δstrap double (Inter FRET) *ATP used | 30 | E140C/ E407C | 1.5 ± 0.01 | | 10.6 ± 0.4 |
| | 25 | E140C/ E407C | 0.92 | | 7.57 |

*Denotes ATP was used for closure. Relates to **Figures 4,6** and **Figure 6—figure supplement 1**. All other reactions used AMPPNP as the ATP analog. Note that 'Intra FRET' in red refers to both probe positions on the same promoter, whereas 'Inter FRET' in blue refers to one probe position per promoter.

## N-terminal residues and kinetic regulation of Hsp90

While the crystal structure of the TRAP1 closed state revealed that the strap made stabilizing interactions with the trans protomer, we show here that its dominant role in modulating ATPase activity is to limit the closure kinetics, presumably though analogous cis-protomer interactions in the apo state. Removal of the strap leads to a ~30-fold increase in ATPase rate and faster closure kinetics that include the smaller conformational steps of NTD-rotation and lid closure, as well as loss of thermal regulation of dimer closure.

The strap extension in TRAP1 appears to continue and expand upon the kinetic regulatory affects observed previously for the first eight residues in yHsp90, which makes contacts on the trans-protomer in the closed state (**Ali et al., 2006**; **Lavery et al., 2014**). Deletion of these residues was shown to accelerate the ATPase rate by ~1.5-fold by allowing H1 and the lid to undergo conformational changes necessary to form trans-protomer contacts at the NTD-dimer interface (**Richter et al., 2002**, **2006**). These effects are understood in the light of numerous apo NTD structures showing this strand makes

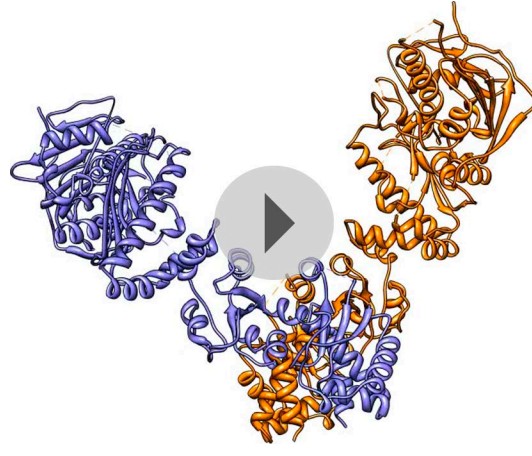

**Video 2**. NTD-strap anti-correlated lid conformational changes. A morph between two conformations of Hsp90, from the Apo state with cis-protomer interactions between NTD and strap, to the nucleotide bound closed state where the strap makes trans-protomer interactions. This morph demonstrates the significant number of contacts that are lost and then reformed to accommodate movement of the NTD to form the NTD-dimerization interface. (PDB codes 2IOQ, 4IVG, 4IPE).

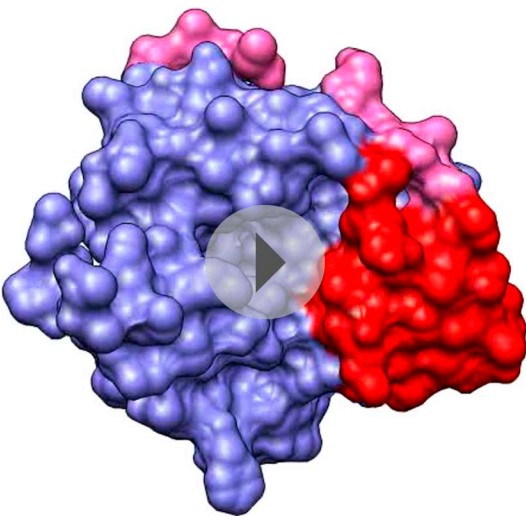

**Video 3**. Movement of the lid to accommodate the NTD-dimerization interface. A morph between two conformations of yHsp90 NTD, either in the APO state or nucleotide bound closed conformation. This morph demonstrates the coordinated movement and changing contacts between both the β-strand (pink) and NTD lid (red) to facilitate the NTD-dimerization interface of a dimerized Hsp90 molecule. (PDB codes 4AS9, 2CG9).

analogous contacts with its own NTD in the apo state (**Shiau et al., 2006**; **Dollins et al., 2007**; **Li et al., 2012**). In the TRAP1 closed-state structure, the 14 ordered residues wrap around the side of the NTD and add an additional 757 Å$^2$, as calculated with PISA (**Krissinel and Henrick, 2007**), of buried surface area and several new trans-protomer contacts (**Lavery et al., 2014**). We propose that similar additional contacts are made in the apo state (**Videos 2 and 3**), which is supported by our own data showing that truncations up the first major contact (salt bridge) have no effect on ATPase (**Figure 3**). Given the similar accelerating effects on ATPase and dynamics as studied in multiple organisms, it is likely that the β-strand and the strap are acting on the same barrier.

Distilling the available information, we outline a model that defines kinetic steps in the Hsp90 ATPase cycle and consequently determine the rates of ATP hydrolysis (**Figure 8A**). Specifically, after ATP is bound, release of cis contacts of the β-strand/strap is coupled to lid closure and NTD rotation, presenting surfaces that form and stabilize an NTD-dimerized state. Due to closure-induced strain this ultimately results in an asymmetric conformation (**Lavery et al., 2014**). Hydrolysis of one of the two ATPs leads to rearrangement of client binding residues (red) between the MD:CTD thus coupling the first ATP to client remodeling when clients are bound to this region. The actual conformational state post hydrolysis of the first ATP is currently unknown, but is here schematized as the symmetric state identified in the yHsp90 crystal structure (**Ali et al., 2006**). After the second ATP is hydrolyzed the chaperone assumes the previously observed compact ADP conformation before resetting the cycle to the apo state.

Specific regulation of the energetic landscape imparted by the TRAP1 strap is depicted in **Figure 8B**. Here, we propose the effect of the strap ultimately impacts the kinetic barrier height as the strap stabilizes both the apo and closed states, although the apo state stabilization is dominant. Thus, addition of a structural element that makes analogous interactions in both the apo (cis) and closed (trans) states provides a novel strategy for kinetic regulation by accentuating the barrier between the apo and closed conformations.

## Functional implications for the evolution of an N-terminal strap

While Hsp90 is very highly conserved across species, there are several regions such as the N-terminus, the charge linker and the very C-terminus that have diverged significantly during evolution. As highlighted in **Figure 7**, the different classes of Hsp90s segregate quite clearly according to

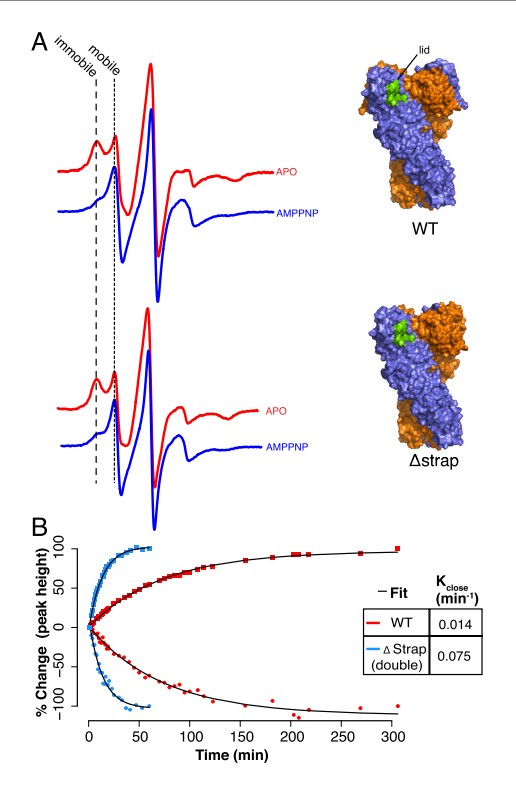

**Figure 5**. Lid Closure rate is regulated by the NTD-strap. (**A**) Continuous Wave (CW) EPR scans of cysteine Free WT (top) and Δstrap zTRAP1 (bottom) labeled with a spin-probe on the NTD-lid (green) in order to observe changes to the lid in the apo and closed states ('Materials and methods'). In the apo state the lid probe shows signal for both mobile and immobile states, although crystallographic data indicate that even in the mobile state, the majority of the lid is still reasonably well ordered. After addition of AMPPNP, the observed signal shifts indicating a predominantly mobile state of the lid, which corresponds to changes in lid dynamics that accompany NTD rotation and dimerization. Only subtle differences are seen in the mobile:immobile peak ratio upon strap deletion. (**B**) CW-EPR scans at ~23°C taken for the cysteine-free WT (red) and Δstrap zTRAP1 (blue) over time after addition of AMPPNP. The percent change in peak height (final vs start) over time is plotted for both the immobile (squares) and mobile (circles) components, showing a clear anti-correlation. The mobile and immobile populations were jointly fit with a single exponential process ('Materials and methods') having a rate constant of 0.014 min⁻¹ for WT and 0.075 min⁻¹ for Δstrap, demonstrating a strong coupling between the strap and the NTD-lid.

the length of their N-termini, with the bacterial and yeast Hsp90s being the shortest, followed in turn by the metazoan cytosolic Hsp90s, the mitochondrial TRAP1s, and the ER Grp94s. One exception is a recently discovered Hsp90α alternative splice variant that creates a very large N-terminal extension of 122 residues. In keeping with observations here, biochemical analysis revealed that this extension is a negative regulator of ATPase activity (*Tripathi and Obermann, 2013*).

In the TRAP1 family, conservation of the strap is strong through the known structured region (His87 in zTRAP1), but decreases towards the N-terminus, and is greatly reduced for TRAP1s from blood fluke, insects, and the sea urchin. Cytosolic Hsp90 has the same drop in conservation and a much shorter strap. By contrast, Grp94 has a very long and very well conserved strap region, with a somewhat variable, but very acidic N-terminus. Despite its long size, deleting the analogous strap region in Grp94 accelerates ATP hydrolysis by only fivefold, although temperature modulation was not investigated (*Dollins et al., 2007*). However, its extreme length, the strong conservation, and the modest effect of deletion on ATPase rates, suggest a possible regulatory role that could couple other phenomena beyond temperature to the rate-limiting conformational changes required for ATP hydrolysis.

The observation that the catalytic efficiency of different Hsp90s vary by ~15-fold (*Richter et al., 2008*) suggest that regulation of the rate-limiting step has been highly tuned through evolution for functional importance. In support of this, yHsp90 mutations that accelerate or decelerate ATPase rates result in significant growth defects and loss of client protein folding in vivo (*Nathan and Lindquist, 1995*; *Prodromou et al., 2000*). The evolution of additional residues at the N-terminus of the Hsp90 gene provides a convenient way to adapt the chaperone's conformational cycle to function with diverse clients encountered by the different homologs or under stressed environmental conditions. Additionally, while the cytosolic Hsp90s are highly regulated by several co-chaperones (*Zuehlke and Johnson, 2010*), only one co-chaperone has been identified for the organellar homologs (*Liu et al., 2010*). This brings forth the possibility that the more extended strap in these homologs could directly or indirectly perform some of the regulation that co-chaperones provide to Hsp90 in the cytosol.

The marked temperature sensitivity observed with TRAP1 raises the intriguing possibility that it represents a homeostatic response in mitochondria where heat is generated through uncoupling of the electron transport chain (*Rousset et al., 2004*). In keeping with the physiological relevance, we demonstrate that the thermal sensitive kinetic barrier is measurably different between zebrafish

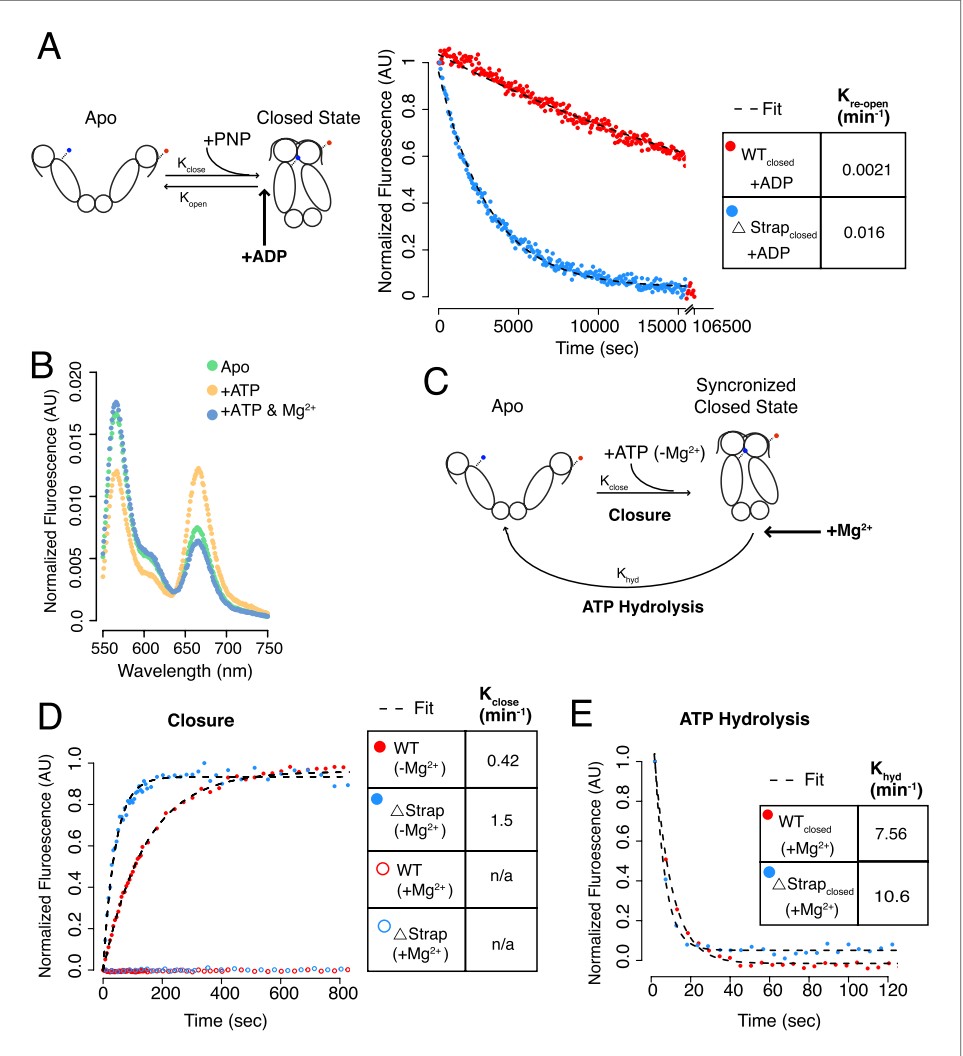

**Figure 6**. The NTD-strap plays a smaller role in additional steps of the ATPase cycle. (**A**) Schematic of dimer closure and re-opening upon addition of AMPPNP (PNP) using the dimer closure FRET probe (left). Re-opening of WT hTRAP1 and Δstrap was induced by 20-fold excess ADP after closure with AMPPNP. Re-opening was accelerated by ~eightfold upon removal of the strap as determined by the ratio of the rates (table inset). (**B**) Steady-state FRET scans of dimer closure FRET in apo and plus ATP in the absence of $Mg^{2+}$. Without $Mg^{2+}$ a closed state accumulates, whereas subsequent addition of $Mg^{2+}$ ('+ATP & $Mg^{2+}$') allows hydrolysis to proceed thereby shifting the population to the apo state. (**C**) Schematic of a kinetic experiment using the $Mg^{2+}$ dependence to separate the rate of hydrolysis from rate of closure. By omitting $Mg^{2+}$, the population can be synchronized in a closed state that is unable to hydrolyze ATP. Subsequent rapid addition of $Mg^{2+}$ leads to ATP hydrolysis, which has now been decoupled from the closure step. (**D**) Kinetic experiments measuring closure and (**E**) ATP hydrolysis. No closed state accumulates if $Mg^{2+}$ is included in the closure reaction. Again we observe that removal of strap residues leads to an accelerated closure rate, whereas the difference in ATP hydrolysis is small. Kinetic rates for each are listed in the table insets.

The following figure supplement is available for figure 6:

**Figure supplement 1**. ATP Hydrolysis at 25°C.

and human TRAP1, which have significantly different physiological temperatures and environments. Additionally, added contacts that the strap provides could be a target for post-translation modifications or even provide a novel binding site for ions, metabolites, or other factors that could modulate the regulatory functions of this element. These observations provide an example of how evolved extensions at the Hsp90 N-terminus can be used to fine-tune chaperone activity to

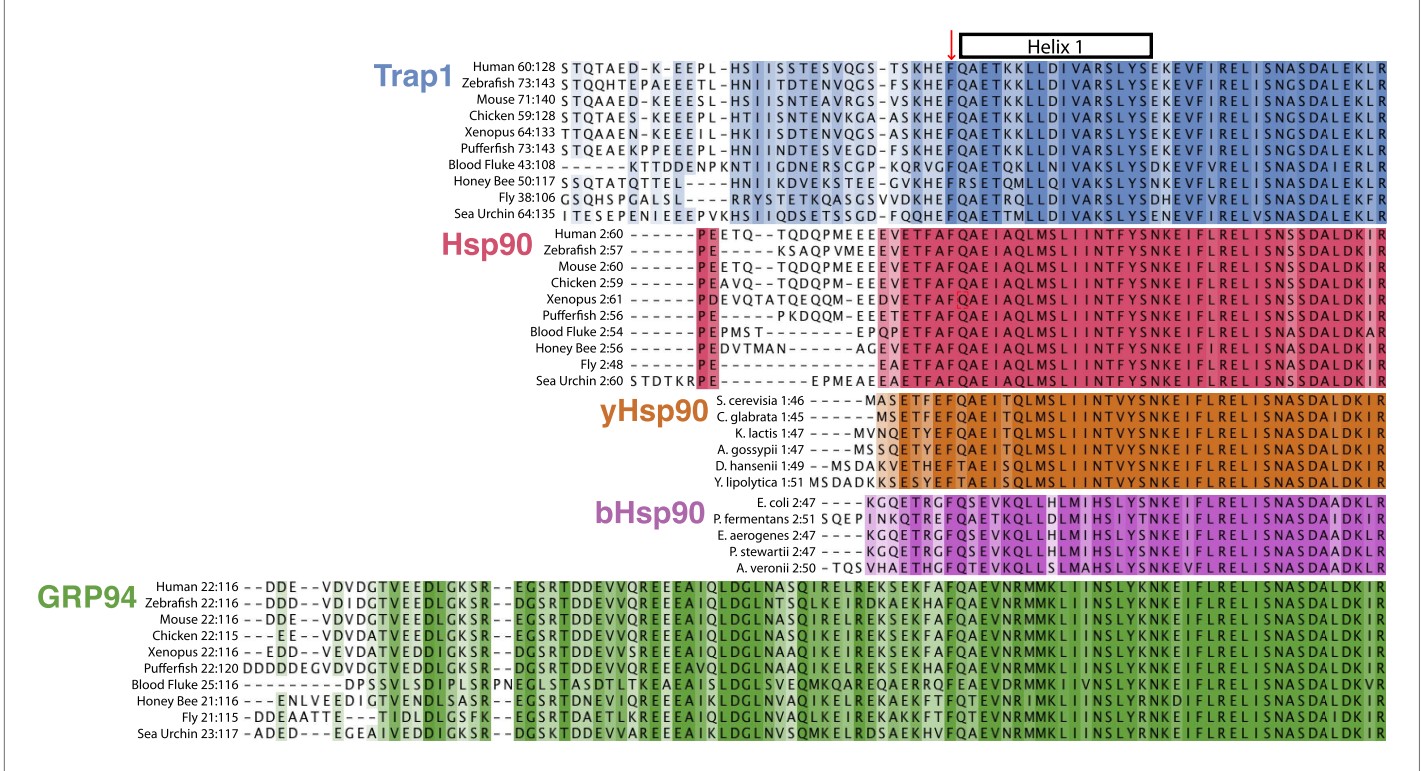

**Figure 7**. Evolution of Hsp90 NTD-strap sequences. Alignments were generated individually for each Hsp90 isoform using a conserved portion of the N-terminal domain and the NTD-strap region. The variable signal sequences for TRAP1 and Grp94 were removed before aligning the 10 divergent sequences. Helix one (H1) of the NTD is annotated above the alignments and begins just after the strictly conserved Phe residue that structurally appears to separate the β-strand region of the NTD from H1. This alignment clearly shows the divergence of both length and sequence within the NTD-strap region and also reveals that residues are more conserved amongst Hsp90 isoforms within H1 and the region following H1. TRAP1 has a much longer strap region than cytosolic Hsp90 and conservation does not pick up until the structural region, as made evident in the TRAP1 crystal structure (*Lavery et al., 2014*). Both yHsp90 and bHsp90 lack a significant strap sequence and Grp94 clearly has an extended and well-conserved strap region.

match organism-specific environmental conditions or unique subcellular demands required for optimal function.

# Materials and methods

## Protein production and purification

Full-length and mutant versions of TNF receptor-associated protein 1 (TRAP1) from *Homo sapiens* and *Danio rerio* (hTRAP1 and zTRAP1, respectively) were purified using our previously described protocol (*Lavery et al., 2014*). The coding sequence of proteins used in this study were cloned into the pET151/D-TOPO bacterial expression plasmid (Life Technologies, Grand Island, NY) and mutant versions of were generated by standard PCR based methods. Cysteine-free hTRAP1 with encoded cysteine positions (Glu140Cys or Glu407Cys) on each or (Ser133Cys and Glu407Cys) on a single protomer, allowed for site-specific labeling with maleimide derivative Alexa Fluor 555/647 dyes (Life Technologies) for FRET experiments. These constructs were also purified as previously described (*Lavery et al., 2014*),with a final size exclusion chromatography storage buffer of 50 mM Hepes pH 7.5, 100 mM KCl, 500 μM TCEP. Aliquots of stored protein were labeled with fluorescent dyes as described below.

## Negative-stain electron microscopy

WT hTRAP1 was initially diluted to 0.1 mg/ml in a buffer containing 20 mM NaH$_2$PO$_4$ pH 7, 50 mM KCl, and 2 mM MgCl$_2$, 0.02% n-octyl-β-D-glucoside + 2 mM AMPPNP. Reactions were incubated at various temperatures for 1 hr (or overnight), followed by dilution to 0.01 mg/ml in the buffer above including 2 mM AMPPNP to maintain nucleotide concentration. 5 μl of the resulting reactions was then

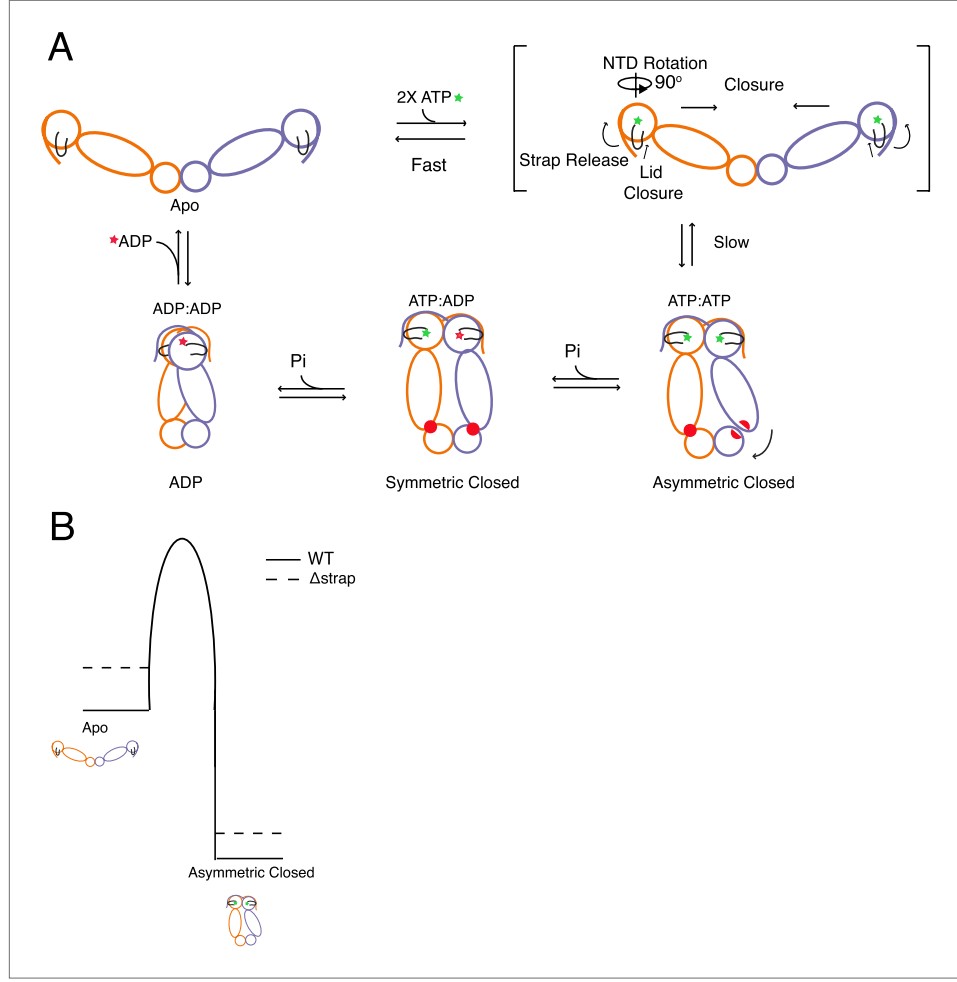

**Figure 8**. Model for the conformational cycle and unique energy landscape of TRAP1. (**A**) In the absence of nucleotide the chaperone is in equilibrium between various open conformations (for simplicity we only show the most open) with the strap folded back onto the cis protomer. Upon binding of ATP, conformational changes necessary for the transition to the closed state are initiated. Here, we propose that the cis contacts of the strap are broken allowing the lid and NTD to undergo conformational changes towards the closed state. After the slow closure step the chaperone assumes the previously reported asymmetric conformation (*Lavery et al., 2014*). Sequential hydrolysis leads to changes in symmetry rearranging the unique MD:CTD interfaces and client binding residues (red) before sampling the ADP conformation and resetting the cycle to the apo state equilibrium. (**B**) Model for the unique energy landscape of TRAP1. Solid lines illustrate the energy landscape of WT TRAP1, and the dashed lines depict the change in landscape upon the loss of the extended N-terminal strap sequence in TRAP1. By stabilizing both the apo and closed states, the strap increases the effective height of the energy barrier. This modulates the conformational landscape, and in the case of hTRAP1 provides pronounced temperature sensitivity.

incubated for ~1 min on 400 mesh Cu grids (Pelco, Redding, CA) coated with a thin carbon layer (~50–100 Å). Following sample incubation, the grid was washed 3× with miliQ water, and lastly stained 3× with uranyl formate pH 6. The final stain was removed by vacuum until the surface of the grid was dry. Prepared grids were imaged with a TECNAI 12 (FEI, Hillsboro, OR) operated at 120 kV. Images were recorded using a 4k × 4k CCD camera (Gatan, Pleasanton, CA) at 52,000 magnification, at −1.5 µm defocus. Representative closed state particles were selected in EMAN (*Ludtke et al., 1999*).

## SAXS data collection and analysis
TRAP1 homologs and mutant proteins were buffer exchanged into 20 mM Hepes pH 7.5, 50 mM KCl, 2 mM MgCl₂, 1 mM DTT. 75 µM protein (monomer concentration) was used as the final concentration for all reactions, and 2 mM AMPPNP was added to initiate closure. Reactions were incubated at

various temperatures for 1 hr followed by a spin at max speed in a tabletop centrifuge for 10 min immediately prior to data collection to remove any trace aggregation.

Data were collected at the Advanced Light Source (ALS) at beamline 12.3.1 with sequential exposure times of 0.5, 1, and 0.5 s. Each sample collected was subsequently buffer subtracted and time points were averaged using scripts provided at beamline 12.3.1 and our own in-house software 'saxs_multiavg.py'. The scattering data were transformed to P(r) vs r using the program GNOM (*Svergun, 1992*) and Dmax was optimized. The resulting distributions were fit using an in-house least squares fitting program 'saxs_combine.py' in the region where non-zero data were present for the target data and closed state model. For the fitting we chose theoretical scattering data for our TRAP1 closed-state model (*Lavery et al., 2014*) and the WT apo data for each TRAP1 homolog. The WT apo data were chosen as the best representation of apo for two reasons. (1) The apo state of Hsp90 proteins consist of a mix of conformations (*Southworth and Agard, 2008*) of which the various conformations and percent of each remains to be elucidated for TRAP1, and (2) removal of the strap (particularly in hTRAP1) induces a shift of the apo distribution towards the closed state as observed for hTRAP1 by SAXS (data not shown), which would result in a value of percent closed for the Δstrap protein that would under represent the true value relative to WT. The theoretical scattering curve for the TRAP1 crystal structure was generated in the program CRYSOL (*Svergun et al., 1995*). The percent of components utilized in the fit and an R factor (R_merge) that is similar to a crystallography R factor in nature is output from our least-squares fitting program and values reported in *Table 1*. R_merge is defined as the equation below

$$R\_merge = \Sigma \, ||Pobs(r)| - |Pcalc(r)| / |Pobs(r)||,$$

where Pobs(r) is the observed probability distribution and Pcalc(r) is the calculated modeled fit. Both pieces of in-house software used for SAXS data analysis, 'saxs_multiavg.py' and 'saxs_combine.py', have been deposited at GitHub.com (https://github.com/agardd/saxs_codes).

## Steady-state ATPase measurements

Steady-state kinetic measurements for various Hsp90 homolog and mutants were carried out in previously described conditions unless otherwise indicated (*Lavery et al., 2014*). Specific buffer conditions used to measure kinetic rates for cysteine free zTRAP1 proteins used in EPR were 20 mM Hepes pH 7.4, 150 mM NaCl, 2 mM MgCl$_2$ at 23°C with 2 mM ATP (see EPR method description). Buffer conditions used to measure kinetic rates for cysteine free hTRAP1 (WT and ΔStrap) used in FRET experiments were 50 mM Hepes pH 7.5, 50 mM KCl, 5 mM MgCl$_2$ with 2 mM ATP (see FRET method description) measured at 30°C. Results were plotted using the program R (*R Development Core Team, 2010*).

## Fluorescence Resonance Energy Transfer (FRET) measurements

Purified protein was labeled with maleimide derivative AlexaFluor 555 (Donor) and 647 (Acceptor) (Life Technologies) at fivefold excess over protein (pre-mixed at 2.5-fold concentration each dye for dual labeled sample) overnight at 4°C. Labeling reactions were then quenched with twofold β-mercaptoethanol over dye concentration and free dye was removed with desalting columns containing Sephadex G-50 resin (illustra Nick Columns, GE Healthcare, Pittsburgh, PA).

For FRET measurements using probes that monitor closure across the dimer (Glu140Cys, Glu407Cys, 'Inter FRET'), labeled protein was mixed at a 1:1 ratio with a final concentration of 250 nM. For measurements with the probe that measures NTD rotation (Ser133Cys and Glu407Cys, 'Intra FRET'), WT hTRAP1 was mixed in 20-fold excess over labeled protein (250 nM labeled protein:5 μM WT). Heterodimers for experiments with all FRET probes were formed at 30°C for 30 min in a reaction buffer consisting of 50 mM Hepes pH 7.5, 50 mM KCl, 5 mM MgCl$_2$. Following heterodimer formation, closure was initiated by addition of 2 mM AMPPNP at various temperatures (*Figure 4*). To measure re-opening, 40 mM ADP was rapidly mixed with pre-closed reactions (closed as in *Figure 4*). For ATP hydrolysis experiments, closure was initiated with 2 mM ATP in a reaction buffer consisting of 50 mM Hepes pH 7.5, 50 mM KCl. After closure was complete, hydrolysis was initiated by rapid addition of 5 mM MgCl$_2$.

Closure and ATP hydrolysis experiments (*Figures 4 and 6B*) were carried out using a Jobin Yvon fluorometer with excitation and emission monochromator slits set to 2 nm/3 nm (respectively), an integration time of 0.3 s, and excitation/emission wavelengths of 532/567 nm (donor) and 532/667 nm

(acceptor). Re-opening experiments (*Figure 6A*) was measured at 30°C using a SpectraMax5 plate reader with excitation and emission wavelengths as above and with a 540 nm emission cutoff. Kinetic measurements were taken at a time interval to minimize photobleaching.

The change in FRET (ratio of Donor and Acceptor fluorescence—division done to graph positive changes and normalized for visual comparison) was well fit with a single exponential (fit in KaleidaGraph, Synergy Software, Reading, PA) to obtain the rate of closure and NTD rotation (Fit 1), as well as re-opening and ATP hydrolysis rates (Fit 2).

$$Fit\,1 : m1 + m2 * \left(1 - exp * \left(-m3 * x\right)\right)$$

$$Fit\,2 : m1 + m2 * \left(exp * \left(-m3 * x\right)\right),$$

where m1 is the time zero value, m2 is the amplitude, m3 is the rate constant, and x is time in seconds. For steady-state FRET scans (taken before and after kinetic measurements), reactions were excited at 532 nm and emission was collected from 550–750 nm. FRET scans were normalized such that the area under the curve is 1.

The activation energy was calculated by fitting a plot of the natural log (ln) of the observed closure rate (y-axis) verses inverse temperature (x-axis) using the equation below (Fit 3)

$$Fit\,3 : ln(k) = -E_a / RT + ln(A),$$

where $E_a$ is the activation energy, T is temperature (kelvin, K), R is the gas constant (kcal K$^{-1}$ mol$^{-1}$), and A is a pre-exponential factor. ATPase rates used to calculate $E_a$ for Hsp90 homologs were taken from reference (*Frey et al., 2007*).

## Continuous-wave electron paramagnetic resonance (EPR)

Cysteine-free zTRAP1 with a Ala201Cys mutation on the lid was exchanged into non-reducing EPR buffer (20 mM Hepes pH 7.4, 150 mM NaCl) at 100 μM (monomer concentration) and labeled by the addition of N-(1-oxyl- 2,2,6,6-tetramethyl-4-piperidinyl)maleimide (MSL, Sigma, St. Louis, MO) to 2.5× concentration of protein overnight at 4°C. The labeled protein was then run through a Micro Bio-Spin column P-30 (Bio-Rad, Hercules, CA) to eliminate free probe. EPR spectra were obtained at ~100 μM labeled protein ± 2 mM AMPPNP in the buffer above with addition of 2 mM MgCl$_2$ and after heating at 30°C for 30 min to ensure closure has completed (*Figure 5A*). For the time course (*Figure 5B*), protein (apo) was spiked with 2 mM AMPPNP and EPR scans recorded overtime at room temperature (~23°C).

EPR measurements were performed with a Bruker EMX EPR spectrometer (Bruker, Billerica, MA) in a 50-μl glass capillary. First derivative X-band spectra were recorded in a high-sensitivity microwave cavity using 50-s, 10-mT-wide magnetic field sweeps. The instrument settings were as follows: microwave power, 25 mW; time constant, 164 ms; frequency, 9.83 GHz; modulation, 0.1 mT at a frequency of 100 kHz. Each spectrum used in the steady-state data analysis was an average of 10–20 sweeps from an individual experimental preparation, with one sweep used for kinetic measurements.

Analysis of the raw peak heights indicated that both the mobile and immobile fractions were changing as a concerted single exponential process. As a consequence, to determine the rate constant, it was unnecessary to account for peak overlaps or the starting fraction in each state. To quantify, the raw peak heights at each time point were determined using the Bruker EMX EPR spectrometer software (Bruker, Billerica, MA) and converted to a percent change over the time course. The rates of lid closure for WT and Δstrap were estimated by fitting the normalized peak heights for each sample to a single exponential decay process with the same rate constant for the mobile and immobile peaks (done as a constrained non-linear fit in Prism v6, GraphPad software, La Jolla, CA).

## Acknowledgements

We would like to thank G Hura, K Dyer, J Tanamachi, and staff of beamline 12.3.1 at the Advance Light Source (ALS) for SAXS data collection and helpful discussions. We also thank D Southworth for assistance with negative-stain EM, as well as numerous members of the Agard lab for helpful discussions. Support for this work was provided by the PSI-Biology grant U01 GM098254 (DAA), HHMI and the Larry L Hillblom Center for the Biology of Aging (LAL).

## Additional information

### Funding

| Funder | Grant reference number | Author |
|---|---|---|
| Howard Hughes Medical Institute | | James R Partridge, Laura A Lavery, David A Agard |
| National Institute of General Medical Sciences | U01 GM098254 | James R Partridge, Laura A Lavery, David A Agard |
| Larry L. Hillblom Foundation | | Laura A Lavery |

The funders had no role in study design, data collection and interpretation, or the decision to submit the work for publication.

### Author contributions

JRP, LAL, DE, DAA, Conception and design, Acquisition of data, Analysis and interpretation of data, Drafting or revising the article, Contributed unpublished essential data or reagents; NN, Acquisition of data, Analysis and interpretation of data, Drafting or revising the article, Contributed unpublished essential data or reagents; RC, Analysis and interpretation of data, Drafting or revising the article, Contributed unpublished essential data or reagents

## Additional files

### Major datasets

The following previously published datasets were used:

| Author(s) | Year | Dataset title | Dataset ID and/or URL | Database, license, and accessibility information |
|---|---|---|---|---|
| Dollins DE, Warren JJ, Immormino RM, Gewirth DT | 2007 | Structure of full length GRP94 with ADP bound | 2O1V; http://dx.doi.org/10.2210/pdb2o1v/pdb | Publicly available at RCSB Protein Data Bank (http://www.rcsb.org/). |
| Menssen R, Orth P, Ziegler A, Saenger W | 2003 | Complex recognition of the supertypic BW6-determinant on HLA-B and-C molecules by the monoclonal antibody SFR8-B6 | 1CG9; http://dx.doi.org/10.2210/pdb1cg9/pdb | Publicly available at RCSB Protein Data Bank (http://www.rcsb.org/). |
| Shiau AK, Harris SF, Southworth DR, Agard DA | 2006 | Crystal Structure of Full-length HtpG, the Escherichia coli Hsp90, Bound to ADP | 2IOP; http://dx.doi.org/10.2210/pdb2iop/pdb | Publicly available at RCSB Protein Data Bank (http://www.rcsb.org/). |
| Lavery LA, Partridge JR, Ramelot TA, Elnatan D, Kennedy MA, Agard DA | 2014 | Crystal structure of mitochondrial Hsp90 (TRAP1) with AMPPNP | 4IPE; http://dx.doi.org/10.2210/pdb4ipe/pdb | Publicly available at RCSB Protein Data Bank (http://www.rcsb.org/). |
| Lavery LA, Partridge JR, Ramelot TA, Elnatan D, Kennedy MA, Agard DA | 2014 | Crystal structure of mitochondrial Hsp90 (TRAP1) NTD-Middle domain dimer with AMPPNP | 4IVG; http://dx.doi.org/10.2210/pdb4ivg/pdb | Publicly available at RCSB Protein Data Bank (http://www.rcsb.org/). |
| Kitson RRA, Chang C, Xiong R, Williams HEL, Davis AL, Lewis W, Dehn DL, Siegel D, Roe SM, Prodromou C, Ross D, Moody CJ | 2013 | The structure of modified benzoquinone ansamycins bound to yeast N- terminal Hsp90 | 4AS9; http://dx.doi.org/10.2210/pdb4as9/pdb | Publicly available at RCSB Protein Data Bank (http://www.rcsb.org/). |
| Ali MMU, Roe SM, Vaughan C, Meyer P, Panaretou B, Piper PW, Prodromou C, Pearl LH | 2006 | Crystal structure of an HSP90-SBA1 closed chaperone complex | 2CG9; http://dx.doi.org/10.2210/pdb2cg9/pdb | Publicly available at RCSB Protein Data Bank (http://www.rcsb.org/). |

| Shiau AK, Harris SF, Southworth DR, Agard DA | 2006 | Crystal Structure of full-length HTPG, the Escherichia coli HSP90 | 2IOQ; http://dx.doi.org/10.2210/pdb2ioq/pdb | Publicly available at RCSB Protein Data Bank (http://www.rcsb.org/). |

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
