## [Decision Letter]

Thank you for sending your work entitled “A novel N-terminal extension in
mitochondrial Hsp90 (TRAP1) serves as a thermal regulator of chaperone activity”
for consideration at *eLife*. Your article has been favorably evaluated
by John Kuriyan (Senior editor), a Reviewing editor, and 2 reviewers.

The Reviewing editor and the reviewers have commented positively on your manuscript.
Both reviewers have identified one point that needs additional support. This issue
concerns the fact that the physiological relevance of the work is not so clear, since
the major effects for human TRAP1 happen at temperatures below 37°C. Both reviewers
were interested in seeing data recorded at higher temperatures. Please address this
important issue in the revised manuscript and as many of the comments in the two reviews
as possible.

Reviewer #1

Hsp90 is a molecular chaperone found in prokaryotes and in the cytosol as well as
organelles of the eukaryotic cell. Its function is coupled to ATP hydrolysis along with
associated large conformational changes. While the key steps of the Hsp90 ATPase cycle
are understood, the regulation of these structural rearrangements is still elusive.
Previously, the authors solved the crystal structure of mitochondrial Hsp90, TRAP1,
bound to AMPPNP.

They identified a 14-residue extension of the N-terminal beta-strand which crosses over
between protomers in the closed state. This 'strap' is found in higher
eukaryotes but is absent in yeast and bacteria. The authors also showed that point
mutations or the deletion of the strap in TRAP1 (of zebra fish) or endoplasmic Grp94
results in an increase in ATPase activity. In this study, they show that a
temperature-dependent kinetic barrier limits the conformational changes from the apo to
the closed form of TRAP1. At lower temperature (23°C) TRAP1is predominantly open,
even in the presence of AMPPNP. Local conformational changes associated with lid closure
are a part of the rate limiting step to the closed state and are regulated by the strap
in TRAP1.

This is an interesting observation adding to our understanding of aspects of the
conformational cycle of Hsp90 and species-specific differences.

However, the biological function of the temperature regulation remains unclear to me,
assuming that 37°C may be the resting state of human mitochondria and higher
temperature present during energy generation or under thermal stress conditions. Thus
the regulation should be active at higher temperatures, such as 42°C. Experiments
addressing this issue would be of interest.

Specific points:

1) The authors state that the kinetic barrier for closure is large and unusually
sensitive to temperature changes. Examples should be included to allow for
comparison.

2) The kinetic data should include fits and resulting rate constants (and also which
equation was used) to judge quality of the kinetic model.

3) If the measured conformational change is indeed rate limiting then its temperature
dependence should be the same as that of the ATPase activity measured before. Is this
indeed the case? How is the relationship of the shortened variant?

4) Figure 1: Do other Hsp90 isoforms also show a
similar trend over a temperature range or is this special just for TRAP1?

5) Experiments in Figures 1 and 2 do not show
kinetics, just a shift of equilibria.

6) EM images for delta strap should be included.

7) In an Agard publication from 2008 (Southworth & Agard, Cell 2008) EM images of
HtpG, yeast and human Hsp90 are shown in the apo state, with AMPPNP, and with ADP. The
human Hsp90α is in an open conformation with AMPPNP at 37°C (according to
them TRAP1 is fully closed at 37°C).

8) Figure 2–figure supplement 1: Isn't this the same figure as Figure 2 just without the delta-strap?

9) Is it known that the trans contacts that strap forms in the closed conformation
stabilize the closed conformation?

10) Figure 4: For the NTD:MD rotation, the FRET
probe does not show a significant change in the FRET signal.

11) Figure 4: Why was saturation not reached for
the 42°C sample?

12) Data for the dimer closure FRET construct should be included and it would be
important to see how delta-strap acts in the FRET assay at different temperatures to
compare with the SAXS data.

13) Why was the double strap mutant too unstable for NTD rotation FRET and not for
inter-protomer FRET? Isn't it the same construct just with a different Cys
label?

14) Figure 5: The legend says that in the apo
state the lid probe is in equilibrium between mobile and immobile states. In the Results
section it is stated that the apo form is predominantly immobile.

15) According to the authors, deleting strap compromises the stability of the closed
structure and hence enhances the reopening rate and shifts the equilibrium towards the
open state. If deleting strap shifts the equilibrium towards the open state, why is the
delta-strap construct predominantly closed?

In this context, the authors mention that the effect of deleting the strap on the
opening of the NTD interface is smaller than the effect on the kinetic barrier
corresponding to the release of strap from the apo state. The respective numbers seem to
be missing from Table 3.

16) Are the ATPase activities of the cys-free version of hTRAP1 and zebrafish TRAP1
mutant identical to the respective wild type proteins?

17) Figure 6. The legend says delta strap is
∼7 fold faster; text says ∼8 fold faster.

18) Figure 6: a trace showing that in presence
of ATP without Mg2+, there is no ATPase activity should be added. Not adding
Mg2+ is not necessarily equivalent to not having (ambient) Mg2+ present in the
solution. The ATP induced changes in FRET signals should be measured also in the
presence of EDTA.

19) The authors mention that in the absence of Mg2+, ATP and AMPPNP show pronounced
differences in kinetics of FRET signal, the difference with/without Mg2+ should be
even more pronounced, that is the kinetics of o/c may be substantially faster in the
presence of Mg2+. Measuring the FRET kinetics upon addition of Mg-ATP and Mg-AMPPNP
is crucial to show that ATP induced closing kinetics in absence of Mg2+ are indeed
representative for the ATPase cycle.

20) How does one know that Mg2+ can actually bind to the closed form; and that a
re-open is not necessary for this to happen?

Reviewer #2

In this study, Partridge and colleagues investigate the role of the N-terminal extension
(“strap”) of TRAP1, the mitochondrial Hsp90 isoform, which in the
previously solved crystal structure of the TRAP1 dimer wraps around the N-domain of the
opposite protomer. They characterized the effects of the strap on the dimer closure
kinetics, the rotation of the N-domain relative to the M-domain, the ATP hydrolysis and
movement of the ATP lid (N-domain) using negative stain electron microscopy (EM), small
angle x-ray scattering (SAXS), fluorescence resonance energy transfer (FRET) and
electron paramagnetic resonance (EPR) measurements. They demonstrate that the strap
region is responsible for a temperature-dependent increase in the rate of TRAP1 closure,
as well as the increase in the ATPase activity.

The data presented here is convincing and interesting. The physiological relevance of
the observed phenomenon is not so clear since the major effects for human TRAP1 happen
at temperatures below 37°C. Nevertheless, the story could be published after the
authors addressed the raised issues.

Major comments

1) Figure 1 shows negative stain EM images of
TRAP1 in the presence of AMPPNP pre-incubated at different temperatures. Few
representative samples are picked from each grid to show the transition from open to
closed conformation with the increase in temperature. The authors should quantify the
open and closed structures from a representative square of the electron micrograph.

2) Using SAXS the authors measured AMPPNP-induced transition of human and zebrafish
TRAP1 to the closed conformation between 20 and 36°C (Figure 2). To determine the physiological relevance of their
observations the authors could have measured the AMPPNP induced transition of human
TRAP1 at 37 to 42°C. Does human TRAP1 become more active at heat shock
temperatures?

3) Figure 4: The authors investigate
AMPPNP-induced changes in TRAP1 conformation using FRET. Control experiments with only
the acceptor dye need to be shown especially as the changes in fluorescence for the
NTD:MD rotation seems to be very small. The temperature at which the experiment of Figure 4 has been performed should be mentioned
in the figure legend.

4) The dimer closure FRET experiments have been performed only at 30°C. As the
paper deals with effect of temperature on ATPase rates of TRAP1, it would be very
important to see the change in rate of dimer closure at different temperatures. It would
also be interesting whether the rates of NTD-MD-rotation and dimer closure are similar
to each other.

5) In Table 2 the authors write that the steady
state ATPase rate for human TRAP1 was 0.463 min-1 and in Table 3 the write that the closing rate for human TRAP1 at
30°C was 0.02 min-1. These values do not fit together and contradict the claims of
the authors. The authors should indicate at which temperature the ATPase assays were
performed and correlate the closing rate with the ATP hydrolysis rates to substantiate
their claims. Maybe the authors will have to measure the closing rate upon addition of
ATP instead of AMPPNP. This seems possible since omission of Mg prevents hydrolysis as
the authors have shown.

---

## [Author Response]

*The Reviewing editor and the other reviewers have commented positively on your
manuscript. Both reviewers have identified one point that needs additional support.
This issue concerns the fact that the physiological relevance of the work is not so
clear, since the major effects for human TRAP1 happen at temperatures below
37°C. Both reviewers were interested in seeing data recorded at higher
temperatures. Please address this important issue in the revised manuscript and as
many of the comments in the two reviews as possible*.

The major question echoed by all was whether the experiments done in this paper could be
done at higher temperatures to better reflect how the observed temperature dependent
activity of TRAP1 would be beneficial above homeostatic temperatures of the organism for
the homologs tested. We have attempted to address these concerns by including new data
recorded at temperatures above 37°C. 37°C data were already included in the
original manuscript. This includes new electron micrographs of samples incubated at
42°C, SAXS data up to 43°C, and better explanation of previously recorded data
demonstrating that the rate of ATPase activity in TRAP1 will continue to increase until
60°C, at which point TRAP1 begins to denature. The new data taken at temperatures
above 37°C has been added to already existing figures.

The new EM data has been incorporated into Figure 1, panel A. The new SAXS data has been incorporated into Figure 2, panels A, B, C, D. Table 1 has also been updated to include data above 37°C, reflecting the
changes made to Figure 2. We also responded to
the reviewer’s requests by including FRET data for both types of probes, intra
and now inter FRET (Figure 4). The increasing
temperature series for both sets of FRET probes behaves similarly and increases as a
response to temperature with each assay having a comparable fold increase. Further we
added extensive temperature series of closure kinetics measured using the inter FRET
probe between 23 and 42°C for the delta strap variant of human TRAP1. Two new
panels have been added with this data (Figure 4). Importantly this data, together with that previously shown in Figure 4 allows calculation of an Arrhenius
activation energy ±strap. This shows that the strap contributes ∼60% of the
activation energy measured for WT TRAP1 (Figure 4, Figure 4 legend and within the main
text). Further, we have made every attempt to improve the manuscript as suggested.

These edits include changes to main text figures, as well as further edits to the text.
We have additionally pointed out examples of data recorded at temperatures above
37°C that were included in the original submission, such as the FRET experiment
measuring the temperature dependence of NTD rotation. These changes and our responses to
individual comments by reviewers are discussed in more detail below.

Reviewer #1

*Hsp90 is a molecular chaperone found in prokaryotes and in the cytosol as well
as organelles of the eukaryotic cell. Its function is coupled to ATP hydrolysis along
with associated large conformational changes. While the key steps of the Hsp90 ATPase
cycle are understood, the regulation of these structural rearrangements is still
elusive. Previously, the authors solved the crystal structure of mitochondrial Hsp90,
TRAP1, bound to AMPPNP*.

*They identified a 14-residue extension of the N-terminal beta-strand which
crosses over between protomers in the closed state. This 'strap' is found
in higher eukaryotes but is absent in yeast and bacteria. The authors also showed
that point mutations or the deletion of the strap in TRAP1 (of zebra fish) or
endoplasmic Grp94 results in an increase in ATPase activity. In this study, they show
that a temperature-dependent kinetic barrier limits the conformational changes from
the apo to the closed form of TRAP1. At lower temperature (23⁰C) TRAP1is
predominantly open, even in the presence of AMPPNP. Local conformational changes
associated with lid closure are a part of the rate limiting step to the closed state
and are regulated by the strap in TRAP1*.

*This is an interesting observation adding to our understanding of aspects of the
conformational cycle of Hsp90 and species-specific differences*.

*However, the biological function of the temperature regulation remains unclear
to me, assuming that 37⁰C may be the resting state of human mitochondria and
higher temperature present during energy generation or under thermal stress
conditions. Thus the regulation should be active at higher temperatures, such as
42⁰C. Experiments addressing this issue would be of interest*.

We have attempted to address the concern of all reviewers by now including

SAXS and EM data taken at temperatures above 37°C. Both high temperature datasets
agree with the basic observation that a compact or “closed” conformation
dominates the population at equilibrium.

With EM we see that the predominant form is a closed conformation at higher temperatures
as demonstrated in Figure 1. Additionally there
continues to be an increase in the % closed population as measured with SAXS, Figure 2. This increase in the % closed population
in both species has been tabulated in Table 1.
Looking at both Figure 2 and Table 1 you can see that the % closed does
continue to increase beyond 37 °C, although there is one outlier in all this data
and that is WT zTRAP1, which shows a decrease in % closed. Presumably, temperatures
above 37° C are physiologically irrelevant for zebrafish. That said, Δstrap
zTRAP1 did continue to show an increase in the % closed population. Concerning
steady-state ATP hydrolysis measurements at temperatures above 37 °C we also
modified the text to make it more obvious that temperature dependence of ATPase had
previously been characterized for TRAP1 by Johannes Buchner’s lab in manuscripts
referenced in the text.

Our original submitted manuscript did include some FRET measurements taken at 42°C
with WT hTRAP1 showing a dramatic increase in the rate of closure compared with
36°C, Figure 4 and Table 3. A 42°C closure rate of the Δstrap variant is
also included in Figure 4 and Table 3.

*Specific points*:

*1) The authors state that the kinetic barrier for closure is large and unusually
sensitive to temperature changes. Examples should be included to allow for
comparison*.

We thank the reviewer for pointing this out and we have tried to better emphasize this
point in the first section of the Results. hHsp90 has no change while yHsp90 and TRAP1
do have temperature sensitivity, however TRAP1 appears to be more extreme. By collecting
a new series of Δstrap temperature data (Figure 4), we can now calculate an Arrhenius activation energy for both the WT and
Δstrap variants of TRAP1. Arrhenius fits are now included in Figure 4 and with modifications included in the main text. We have
also included a quantification of the activation energy for Hsp90 homologs (yHsp90 and
Grp94) in Figure 4—figure supplement 2,
utilizing previously reported rates found in Frey et al 2007. Comparing our calculated
activation energy as well as a previously reported value (Leskovar et al 2008), we find
that the other homologs have significantly lower activation energies. These data clearly
show that TRAP1 has unusually large response to temperature changes.

*2) The kinetic data should include fits and resulting rate constants (and also
which equation was used) to judge quality of the kinetic model*.

Agreed. The kinetic data in the manuscript now all include a plot of the fit used to
determine the rate constant. The equations are now included in the methods section as
well.

*3) If the measured conformational change is indeed rate limiting then its
temperature dependence should be the same as that of the ATPase activity measured
before*. *Is this indeed the case? How is the relationship of the
shortened variant?*

We have better highlighted in the text our observation of the difference in closure rate
(measured by FRET) with AMPPNP and ATP, and the steady-state ATPase rates. We find that
the closure rate measured with ATP better matches the ATPase rate measured for the fully
labelled Inter FRET probe used in experiments shown in Figure 6 and in the same buffer and temperature conditions (see Methods).
Though we have not measured the temperature dependence of closure with ATP at the full
range of temperatures as AMPPNP, we do show that the closure rate is slower at 25
°C and that the fold difference in closure rate is greater between WT and
Δstrap protein at 25 °C (Table 3).
The difference between the ATP analogs suggests that the energetics differ which could
shift (but not mitigate) the observed temperature dependence of closure depending on the
analog used. Importantly, we point out that despite experimental differences that could
arise due to cysteine removal, labelling, or usage of varying nucleotide, our
observation of the unique temperature dependent closure (also supported by Leskovar et
al 2008) and the role of the strap as a structural element responsible for regulating
this observation remains constant across all experiments in our manuscript.

Our proposal that closure is rate limiting is supported by previous studies (Hessling et
al 2009 and Leskovar et al 2008) as well as our measurements in Figures 4 and 6, which allow us to evaluate the rate of
closure, re-opening and hydrolysis in matching conditions for our WT and Δstrap
FRET probes. We were able to decouple the closure and hydrolysis steps by removing MgCl2
from the closure reactions in the presence of nucleotide and find that closure is much
slower than hydrolysis, with both WT and Δstrap hTRAP1 (Figure 6, expounded upon within our manuscript). These
experiments also show that removal of the strap has the greatest impact on the closure
rate, a significant but smaller effect on re-opening, and a minor effect on
hydrolysis.

We have additionally included FRET experiments with the Inter FRET probes to monitor the
rates of closure ± the strap (Figure 4).
From this data an Arrhenius plot of both the WT and Δstrap proteins has been
added in Figure 4. Calculating the difference in
Ea between WT and Δstrap we assign the contribution of the strap to Ea at
approximately 60% of the measured Ea for WT hTRAP1 (48.8 kcal/mol Ea for WT; 29 kcal/mol
for Δstrap). These data are consistent with the steady-state SAXS and ATPase, and
show that removal of the strap region lowers the energy barrier between apo and the
closed state.

*4)*
Figure 1*: Do other Hsp90
isoforms also show a similar trend over a temperature range or is this special just
for TRAP1?*

This trend is not just specific for TRAP1 and we would like to point the reviewers to
Leskovar et al 2008 and Krukenberg et al 2008. The sensitive temperature range and
specific rates do vary dramatically among Hsp90 homologs, with TRAP1 displaying
particularly heightened sensitivity (also see specific point 1 author response above).
Regulation via the strap is the focus of our manuscript and the strap does not exist in
the yHsp90 and bHsp90 homologs. In addition to TRAP1 and Grp94, the temperature
sensitivity has also been recently reported for an Hsp90α alternative splice
variant (Tripathi et al 2013) that importantly is imparted by a long N-terminal
extension of ∼122 amino acids. We have made an effort to highlight these points
in our discussion section.

*5) Experiments in*
Figures 1 and 2
*do not show kinetics, just a shift of equilibria*.

We have changed the titles and Figure descriptions to better highlight these experiments
as equilibrium experiments. These experiments initially suggested to us that TRAP1 might
have a unique energy landscape, which we set out to elucidate the underlying mechanism
of the phenomena.

*6) EM images for delta strap should be included*.

We have not collected EM images for Δstrap. After taking the initial EM images of
the WT at various temperatures we moved to measure the % closed state by SAXS, which is
a much more quantitative measure of conformational states and has previously be used to
measure Hsp90 conformational equilibrium by our lab and others (Frey et al 2007).

*7) In an Agard publication from 2008 (Southworth & Agard, Cell 2008) EM
images of HtpG, yeast and human Hsp90 are shown in the apo state, with AMPPNP, and
with ADP. The human Hsp90α is in an open conformation with AMPPNP at
37⁰C (according to them TRAP1 is fully closed at 37⁰C)*.

TRAP1 is the mitochondrial variant that shows noticeably different behavior from the
cytosolic form of hHsp90 that was shown in the (Southworth & Agard, Cell 2008)
manuscript. In this study we sought to bootstrap from our recent TRAP1 crystal structure
(none are available for Hsp90α) to gain possible molecular insights into this
difference. Cytosolic hHsp90 does not significantly close with AMPPNP at 37 °C
suggesting a different energetic landscape and consequently a much lower ATPase
activity, as highlighted in Southworth et al. It is important to note that although the
energetics are different between Hsp90 homologs (likely due to different physiological
environments, specific clients, and different requirements for co-chaperones) the
underlying conformational states and mechanism of protein folding are conserved as also
highlighted in Southworth et al.

*8) Figure 2–figure supplement 1: Isn't this the same figure
as*
Figure 2
*just without the delta-strap?*

Figure 2–figure supplement 1 (now Figure 2) is meant to demonstrate that TRAP1 will remain closed even after cooling
the sample back to 20°C for two hours. By this observation TRAP1 is kinetically
trapped in the closed state after heating and will remain closed even when cooled. These
data support a large temperature dependent barrier to the closed state that is overcome
upon increasing temperature and a stable closed state once the transition has occurred
rather than a pronounced temperature dependence of the equilibrium states. As mentioned
above, the supplemental figure has now been combined with Figure 2, panel D, in an attempt to make this less confusing.

9) Is it known that the trans contacts that strap forms in the closed
conformation stabilize the closed conformation?

From our previously published crystal structure of TRAP1 (Lavery et al 2014),
cocrystallized with AMPPNP, we know that the strap makes substantial contacts with the
trans-NTD while in the closed conformation, suggesting a role in stabilization. However,
this is most directly shown by our new observation that deleting the strap accelerates
reopening (Figure 6).

*10)*
Figure 4*: For the
NTD:MD rotation, the FRET probe does not show a significant change in the FRET
signal*.

The change in FRET signal with the NTD:MD rotation probe is on par with previously
published results using the same probes but with bHsp90 (Street et al, 2011). The
measurements are quite reliable even though the delta for this set of probe positions is
less than that for the cross protomer set.

*11)*
Figure 4*: Why was
saturation not reached for the 42⁰C sample?*

It does. We have now included the full dataset in Figure 4 to better depict saturation at 42°C.

*12) Data for the dimer closure FRET construct should be included and it would be
important to see how delta-strap acts in the FRET assay at different temperatures to
compare with the SAXS data*.

We now show temperature dependent closure as measured with FRET using both the NTD
rotation and dimer closure probes (Figure 4). We
observe that the dimer closure probes (Inter FRET) display comparable temperature
dependent closure as the NTD rotation probe (Intra FRET). Most directly, we have now
included a new temperature dependent closure series for the Δstrap variant (Figure 4).

These measurements show a dramatic loss of temperature dependent closure and
quantification of the activation energy difference (Arrhenius plot using WT and
Δstrap FRET data) shows that the strap contributes over half of the WT Ea (Figure 4).

The FRET data is in good agreement with the SAXS data, which shows an increase in closed
state at higher temperatures measured after 1 hour at the respective temperature. By
estimating the % closed state at 1 hour from the FRET data (Figure 9) we get 9%, 42%, 66%, 90% and 87% (compared
to 2%, 31%, 41%, 74% and 84% at the respective temperatures, Table 1). Considering the 1 hour time point for the Δstrap,
the reaction has reached completion according to our FRET and SAXS measurements in Figures 3 and 4, respectively. Although our
normalized FRET data is only an estimate of the percent closed state molecules and
shouldn’t be taken as a quantitate number, estimated percent closed state
compared to the SAXS data show the same trend.Author response image 1.Adapted from Figure 4 (Intra FRET probe
data set).

13) Why was the double strap mutant too unstable for NTD rotation FRET and not
for inter-protomer FRET? Isn't it the same construct just with a different Cys
label?

No, it is a different construct having two incorporated cysteine’s (S133C.E407C),
within one protomer, whereas the inter-protomer FRET uses only 1 Cys per protomer
(either E407C or E140C). We do not have a detailed explanation for why removing the
strap in combination with the 2 Cys mutations in one protomer with the NTD rotation FRET
pair destabilizes the protein. During purification of the Δstrap NTD FRET protein
most was lost to cleavage products even when protease inhibitors were included.

The small amount of full-length protein that was purified did not appear unstable during
experimental measurements, but given the unusual behavior during purification we did not
have enough confidence in our measurements to report quantitative rates and draw
conclusions from these measurements.

*14)*
Figure 5*: The legend
says that in the apo state the lid probe is in equilibrium between mobile and
immobile states. In the Results section it is stated that the apo form is
predominantly immobile*.

We have reworded the legend to state that the apo state is in equilibrium between mobile
and immobile as measured with EPR. After addition of AMPPNP there is a substantial
change such that the mobile population dominates.

15) According to the authors, deleting strap compromises the stability of the
closed structure and hence enhances the reopening rate and shifts the equilibrium
towards the open state. If deleting strap shifts the equilibrium towards the open
state, why is the delta-strap construct predominantly closed?

*In this context, the authors mention that the effect of deleting the strap on
the opening of the NTD interface is smaller than the effect on the kinetic barrier
corresponding to the release of strap from the apo state. The respective numbers seem
to be missing from*
Table 3.

The respective numbers that should be considered are listed in Table 3 under the Kclose and Kreopen heading for the CysFree hTRAP1
(E140C/E407C) and the Δstrap double (E140C/E407C) constructs. By taking the ratio
of the rates, Kclose has a 16-fold difference while Kreopen has an 8-fold difference.
This is further described in the main text in the FRET based “Dissecting further
regulatory functions of the NTD-strap” section of the Results section in this
manuscript. Thus deleting the strap destabilizes the open state more than it
destabilizes the closed state. However, the overall equilibrium is still in favor of the
open state in the absence of ATP.

16) Are the ATPase activities of the cys-free version of hTRAP1 and zebrafish
TRAP1 mutant identical to the respective wild type proteins?

No. The cys-free versions of hTRAP1 and zTRAP1 are faster than WT in ATPase rates by
1.5–2 fold. This is presented in our previous manuscript (Lavery et al 2014). The
molecular basis for the increased ATPase in the Cysteine Free TRAP1 constructs is
unknown; however, all of our measurements are done by comparing WT to mutant
(Δstrap, or strap point mutants) in matching conditions. We are interested and
drawing conclusions from the fold change between mutant protein and the respective WT
protein. As mentioned above we also point out that despite any differences that come
about due to cysteine removal, labelling or nucleotide used in the experiments, our
observation of the unique temperature dependent closure and the role of the strap as the
structural piece responsible for these observation remains constant across all
experiments in our manuscript.

*17)*
Figure 6*. The legend
says delta strap is ∼7 fold faster; text says ∼8 fold
faster*.

Thank you for pointing this out. This was a mistake and the difference has now been
corrected. The legend now matches the text with “8-fold”.

*18)*
Figure 6*: a trace
showing that in presence of ATP without Mg2+, there is no ATPase activity should
be added. Not adding Mg2+ is not necessarily equivalent to not having (ambient)
Mg2+ present in the solution. The ATP induced changes in FRET signals should be
measured also in the presence of EDTA*.

The coupled NADH reaction used to measure ATPase rates is dependent on

Mg2+ so we are unable to do the comparable ATPase experiment in absence of
Mg2+.

*19) The authors mention that in the absence of Mg2+, ATP and AMPPNP show
pronounced differences in kinetics of FRET signal, the difference with/without
Mg2+ should be even more pronounced, that is the kinetics of o/c may be
substantially faster in the presence of Mg2+. Measuring the FRET kinetics upon
addition of Mg-ATP and Mg-AMPPNP is crucial to show that ATP induced closing kinetics
in absence of Mg2+ are indeed representative for the ATPase cycle*.

While the closure rate with ATP-Mg2+ would be ideal to compare to AMPPNPMg2+
(all FRET data in Figure 4 is done with
AMPPNP-Mg2+), we are unable to measure closure as hydrolysis is faster than
closure, hence the closed state does not build up in the presence of Mg2+. Rather,
we measure a FRET signal that is always equivalent to Apo (no change from time zero).
This was also seen in FRET studies with yeast Hsp90 (yHsp90) in Hessling et al, 2009. We
noted the differences in the manuscript as an observation, however we feel uncovering
the molecular reason for variability of rates and affinity between the two ATP analogs
is beyond the scope of this manuscript. While interesting, here our focus is on the
changes in these rates that are connected with the strap.

20) How does one know that Mg2+ can actually bind to the closed form; and
that a re-open is not necessary for this to happen?

ATP hydrolysis upon addition of Mg2+ is relatively fast compared to the reopening
rate 0.463 vs. 0.002 for WT protein or 13.3 vs. 0.016 for Δstrap protein. This
strongly suggests that Mg2+ can bind to the closed state.

Reviewer #2

*In this study, Partridge and colleagues investigate the role of the N-terminal
extension (“strap”) of TRAP1, the mitochondrial Hsp90 isoform, which in
the previously solved crystal structure of the TRAP1 dimer wraps around the N-domain
of the opposite protomer. They characterized the effects of the strap on the dimer
closure kinetics, the rotation of the N-domain relative to the M-domain, the ATP
hydrolysis and movement of the ATP lid (N-domain) using negative stain electron
microscopy (EM), small angle x-ray scattering (SAXS), fluorescence resonance energy
transfer (FRET) and electron paramagnetic resonance (EPR) measurements. They
demonstrate that the strap region is responsible for a temperature-dependent increase
in the rate of TRAP1 closure, as well as the increase in the ATPase
activity*.

*The data presented here is convincing and interesting. The physiological
relevance of the observed phenomenon is not so clear since the major effects for
human TRAP1 happen at temperatures below 37⁰C. Nevertheless, the story could
be published after the authors addressed the raised issues*.

We thank reviewer for the suggested experiments to strengthen our manuscript. We have
attempted to address this concern, echoed by both the editor and Reviewer 1, by
including new EM and SAXS data recorded above 37°C. It is clear from the SAXS data
that the rate of closure continues to increase as temperatures increase above 37
°C. This is also clear from the FRET closure data (Figure 4) which shows a significant increase in closure rate at 42°C. A
more detailed description of the additional data and references has been described in
the comments above for Reviewer 1.

Major comments

*1)*
Figure 1
*shows negative stain EM images of TRAP1 in the presence of AMPPNP pre-incubated
at different temperatures. Few representative samples are picked from each grid to
show the transition from open to closed conformation with the increase in
temperature. The authors should quantify the open and closed structures from a
representative square of the electron micrograph*.

We very much agree that quantification of the %closed verses Apo state at each
temperature is quite important and must be included. While we have done this in the past
(Southworth, 2008), in practice, this is a somewhat painful and laborious procedure to
do rigorously and instead here we chose to quantify the %closed state by SAXS. This is a
significantly more quantitative assay for measuring equilibrium of states and has
previously be used to measure Hsp90 conformational equilibrium by our lab and others
(Frey et al, 2007). The quantification of %closed state as measured by SAXS can be seen
in Figure 3 and Table 1.

*2) Using SAXS the authors measured AMPPNP-induced transition of human and
zebrafish TRAP1 to the closed conformation between 20 and 36⁰C (*Figure 2*). To determine the
physiological relevance of their observations the authors could have measured the
AMPPNP induced transition of human TRAP1 at 37 to 42⁰C. Does human TRAP1
become more active at heat shock temperatures?*

We now include SAXS data above 37°C and up to 43°C. Interestingly, TRAP1 from
humans seems to have the largest jump in activity around 36-40°C, just where it
might be most physiologically relevant. Analogously, the largest jump in % closed for
TRAP1 from zebrafish seems to be in the 20-30°C range. Both species loose
temperature sensitivity without the strap. However both species do show a jump in
activity in Δstrap when going up to 43°C.

*3)*
Figure 4*: The authors
investigate AMPPNP-induced changes in TRAP1 conformation using FRET. Control
experiments with only the acceptor dye need to be shown especially as the changes in
fluorescence for the NTD:MD rotation seems to be very small. The temperature at which
the experiment of*
Figure 4
*has been performed should be mentioned in the figure legend*.

Addition of nucleotide showed an appropriate anti-correlation of the donor/acceptor FRET
signals and the anticipated direction of the change in FRET (increase in FRET for Inter
FRET probe, and decrease in FRET for Intra FRET probe). The steady-state scans shown in
Figure 4 were taken after the closure
reaction was complete. Here, to avoid any temperature dependence on the dyes, etc.,
closure was induced by heat shock for 1hr and then the samples actually measured at room
temperature. The temperature at which the scans were done has been added to the figure
legend.

*4) The dimer closure FRET experiments have been performed only at 30⁰C.
As the paper deals with effect of temperature on ATPase rates of TRAP1, it would be
very important to see the change in rate of dimer closure at different temperatures.
It would also be interesting whether the rates of NTD-MD-rotation and dimer closure
are similar to each other*.

Because NTD-MD rotation is tightly coupled to closure, the closure rates measured by
either probe set are similar; we had initially hoped to be able to tease apart these
individual steps, but in practice, they seem kinetically inseparable (differences due to
probe locations). We have now included the temperature dependent closure experiment
(Figure 4) and note that both probe sets are
comparable in matching experiments. We chose to do the dimer closure FRET experiment
+/- strap at 30 °C with SAXS, EM and ATPase as this is the temperature where
we see the largest differences between WT and Δstrap in the temperature range
assayed. Additionally, we have now included a temperature dependent closure series for
the Δstrap variant (Figure 4). As
predicted, comparing the fold changes of closure rates (Table 3) at each temperature we see the largest fold change at
lower temperatures (23 °C: 24-fold, 30 °C: 16-fold, 32 °C: 12-fold, 36
°C: 7-fold, and 42 °C: 3-fold). The clear impact of the strap is revealed from
a comparison of the Arrhenius plots in Figure 4.

*5) In*
Table 2
*the authors write that the steady state ATPase rate for human TRAP1 was 0.463
min-1 and in*
Table 3
*the write that the closing rate for human TRAP1 at 30⁰C was 0.02 min-1.
These values do not fit together and contradict the claims of the authors. The
authors should indicate at which temperature the ATPase assays were performed and
correlate the closing rate with the ATP hydrolysis rates to substantiate their
claims. Maybe the authors will have to measure the closing rate upon addition of ATP
instead of AMPPNP. This seems possible since omission of Mg prevents hydrolysis as
the authors have shown*.

We do recognize the significant difference between closure rates determined with AMPPNP
and the ATPase measurements taken with ATP. For reference the best comparison should be
done with the same protein and buffer conditions used in the FRET experiments with ATP
as indicated in Table 2 and Table 3 (0.79 min-1 ATPase- Table 2 vs. 0.42 min-1 closure- Table 3). There is a strong nucleotide dependence on the closure rates, and
the closed state never builds up with ATP plus Mg2+ as hydrolysis is faster than
closure. However, for this manuscript we would like to focus on a more broadened and
molecular aspect of the strap mediating temperature sensitivity to regulate closure and
thereby activity of TRAP1 in both zebrafish and human. We have done matched experiments
for WT and Δstrap for each assay and have drawn our conclusions from the
differences between these matched experiments.

We show the temperature dependence of the closure reaction and the loss of temperature
dependence upon deletion of the strap is robustly observed between multiple biophysical
and biochemical of experiments. We ask that the reviewers please excuse our reluctance
to dive even further into molecular differences between ATP analogs as we feel this is
beyond the scope of our study.